# BubbleML: A Multiphase Multiphysics Dataset and Benchmarks for Machine Learning

**Sheikh Md Shakeel Hassan**[1][*][†]   **Arthur Feeney**[1][*][†]   **Akash Dhruv**[2]   **Jihoon Kim**[3]

**Youngjoon Suh**[1]   **Jaiyoung Ryu**[3]   **Yoonjin Won**[1]   **Aparna Chandramowlishwaran**[1][†]

[1]University of California, Irvine   [2]Argonne National Laboratory   [3]Korea University

## Abstract

In the field of phase change phenomena, the lack of accessible and diverse datasets suitable for machine learning (ML) training poses a significant challenge. Existing experimental datasets are often restricted, with limited availability and sparse ground truth, impeding our understanding of this complex multiphysics phenomena. To bridge this gap, we present the BubbleML dataset [3] which leverages physics-driven simulations to provide accurate ground truth information for various boiling scenarios, encompassing nucleate pool boiling, flow boiling, and sub-cooled boiling. This extensive dataset covers a wide range of parameters, including varying gravity conditions, flow rates, sub-cooling levels, and wall superheat, comprising 79 simulations. BubbleML is validated against experimental observations and trends, establishing it as an invaluable resource for ML research. Furthermore, we showcase its potential to facilitate the exploration of diverse downstream tasks by introducing two benchmarks: (a) optical flow analysis to capture bubble dynamics, and (b) neural PDE solvers for learning temperature and flow dynamics. The BubbleML dataset and its benchmarks aim to catalyze progress in ML-driven research on multiphysics phase change phenomena, providing robust baselines for the development and comparison of state-of-the-art techniques and models.

## 1  Introduction

Phase-change phenomena, such as boiling, involve complex multiphysics processes and dynamics that are not fully understood. The interplay between bubble dynamics and heat transfer performance during boiling presents significant challenges in accurately predicting and modeling these heat and mass transfer processes. Machine learning (ML) has the potential to revolutionize this field, enabling data-driven discovery to unravel new physical insights [1], develop accurate surrogate and predictive models [2, 3], optimize the design of heat transfer systems, and facilitate adaptive real-time monitoring and control [4].

The applications of ML in this domain are diverse and impactful. Consider the context of high-performance computing in data centers, where efficient cooling is critical. Boiling-based cooling techniques, such as two-phase liquid cooling, offer enhanced heat dissipation capabilities, ensuring reliable and optimal operation of power-intensive electronic components such as GPUs [5, 6]. Boiling also plays a central role in optimizing heat transfer in nuclear reactors, where precise modeling and prediction of boiling dynamics contribute to advancing the safety and efficiency of nuclear power systems [7]. Furthermore, boiling processes play a vital role in thermal desalination methods that provide clean drinking water in water-scarce regions [8]. These advancements in pivotal areas such

---

[*]Equal contributions

[†]Corresponding authors: {sheikhh1,afeeney,amowli}@uci.edu

[3]https://github.com/HPCForge/BubbleML

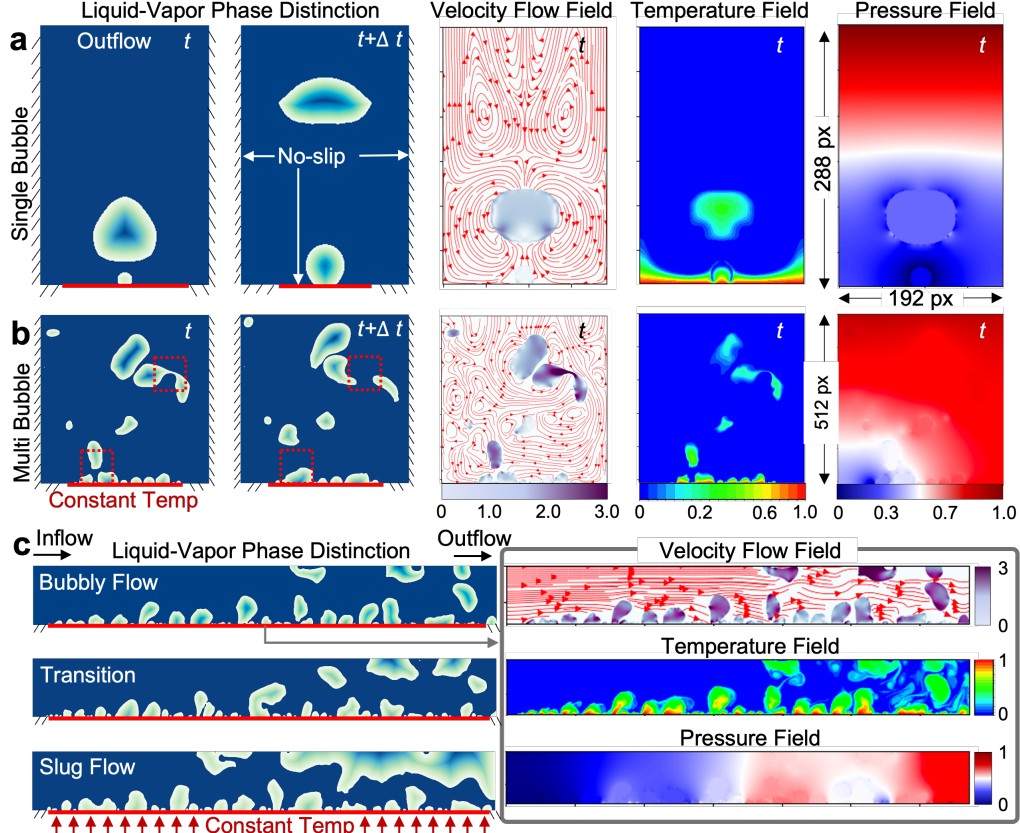

Figure 1: **BubbleML Dataset.** Diverse two-phase boiling phenomena with ground truth for key physical variables–velocity, temperature, and pressure. (a) Single bubble rising from a nucleation site on the heater surface. (b) Chaotic multi-bubble dynamics–merging and splitting. (c) Flow boiling transitions from bubbly to slug regime with increasing inlet velocity. Velocity and temperature fields are obtained by solving equations 1a and 1b, pressure field is obtained by solving the Poisson equation which ensures that continuity is satisfied. The physical quantities are in non-dimensional units.

as thermal management, energy efficiency, and heat transfer applications, driven by ML techniques have far-reaching implications, empowering us to design more sustainable energy systems, enhance environmental preservation efforts, and advance engineering capabilities across various domains.

To train data-driven ML algorithms effectively, we need large, diverse, and accurately labeled datasets. However, obtaining high-fidelity datasets that encompass a wide range of phase-change phenomena and operating conditions is a significant challenge. Boiling processes are highly sensitive to factors like surface properties, pressure, orientation, and working fluid composition [9]. Additionally, the chaotic nature of vapor interactions and occlusions makes quantifying boiling processes inherently difficult. Specialized experimental setups, involving instrumentation, sensors, and high-speed visualization techniques, come with substantial costs, further limiting the availability of extensive and accurate large-scale experimental data [3]. As a result, only a few well-funded research laboratories have access to precise ground truth data, and even then, this data often lacks fidelity and fails to capture detailed microscale dynamics, such as local bubble-induced turbulence and its impact on overall heat transfer. This scarcity of high-fidelity datasets poses challenges in designing accurate ML models for multiphase and phase change processes. While scientific ML (SciML) approaches can incorporate physical knowledge and constraints into the training process to reduce some of this data burden [10], the validation and quantification of uncertainty still rely on the availability of ground truth data. Therefore, there is an urgent need for open, diverse, and large-scale datasets to develop robust models and advance research in multiphysics problems such as phase change phenomena.

Simulations have played a key role as the third pillar of science in overcoming the inherent challenges faced by experimental studies in various scientific domains. High-fidelity multiscale data from simulations complement and enhance experimental measurements. In the field of phase change, simulations

have successfully modeled transport equations for momentum, energy, and phase transition, enabling accurate measurements of velocity, pressure, and temperature fields around bubbles [11, 12]. As a result, simulations serve as powerful tools for understanding and quantifying boiling. However, SciML researchers often set up their own simulations to generate ground truth solutions for training and testing their models rather than relying on open-source benchmark datasets. This practice is common even among impactful papers [10, 13, 14, 15]. While this approach is reasonable for studying specific, simple partial differential equations (PDEs), real-world applications of PDE solvers and simulations often involve large-scale systems with complex multiphase physics and a combination of Dirichlet and Neumann boundary conditions [16]. These real-world problems require significant domain expertise, engineering time, and computational resources. Independently performing such simulations is impractical and even infeasible for many ML researchers. This difficulty in dataset generation has led to a drought of SciML research to study *real-world* physics problems. Prior efforts to build benchmark datasets have primarily focused on single- and multiphysics problems with single-phase dynamics [17, 18, 19, 20].

As a response to the above challenges, we introduce the BubbleML Dataset [4], an extensive and innovative collection of data generated through Flash-X simulations [21]. This dataset encompasses a wide range of boiling phenomena, including nucleate boiling of single bubbles, merging bubbles, flow boiling in different configurations, and subcooled boiling. Figure 1 is a visual glimpse into the diverse range of physical phenomena and variables in the dataset. To further enhance its applicability, the dataset covers various gravity conditions ranging from earth gravity to gravity at the International Space Station, different heater temperatures, and different inlet velocities. In total, we present around 80 simulations, each capturing a specific combination of parameters and conditions. In summary, the key contributions are as follows:

**Multiphase and Multiphysics Dataset.** A comprehensive dataset encompassing a range of two-phase (liquid-vapor) phase change phenomena in boiling, with a focus on bubble and flow dynamics.

**Real-world Validation.** Validation against experimental data to ensure the dataset's accuracy and reliability. This validation process enhances the dataset's fidelity and establishes a strong connection between simulation and real-world phenomena.

**Diverse Downstream Tasks.** BubbleML is designed to facilitate diverse downstream applications. To demonstrate its potential, we present two benchmark tasks: optical flow for learning bubble dynamics and neural PDE solvers for modeling temperature and flow dynamics.

## 2 Related Work

**Scientific Machine Learning Datasets.** There have been several efforts to develop benchmark datasets for scientific machine learning tasks [17, 18, 19, 20, 22, 23]. Notably, the ERA5 atmospheric reanalysis dataset [23], curated by the European Center for Medium-Range Weather Forecasting (ECMWF) provides hourly estimates of a large number of atmospheric, land, and oceanic climate variables since 1940. It is the most popular publicly available source for weather forecasting, facilitating the training of neural weather models such as FourCastNet [24], GraphCast [25], and ClimaX [26]. PDEBench [17] provides an impressive collection of datasets for 11 PDEs commonly encountered in computational fluid dynamics. Boundary conditions in scientific simulations play a crucial role in capturing the dynamics of the underlying physical systems. The majority of datasets in PDEBench utilize periodic boundary conditions. Although some datasets encompass Neumann or Dirichlet boundary conditions, none consider a combination of both which presents a noteworthy gap in accurately modeling real-world scenarios. Another challenging problem is the modeling of turbulent Kolmogorov flows and the dataset generated using JAX-CFD [27] is gaining popularity in benchmarking neural flow models [28, 29]. BlastNet [19] generated using DNS solver, S3D [30] focuses on simulating the behavior of a single fluid phase solving for compressible fluid dynamics, combustion, and heat transfer. AirfRANS [20] is a dataset for studying the 2D incompressible steady-state Reynolds-Averaged Navier–Stokes equations over airfoils. Current datasets have made commendable strides in addressing single- and multiphysics scenarios, and provide a valuable foundation for developing and evaluating SciML algorithms. Nonetheless, their scope falls short of capturing the range of behaviors and phenomena encountered in phase change physics.

---

[4]Through Zenodo, a permanent DOI for the dataset 10.5281/zenodo.8039786

In contrast, BubbleML focuses on capturing the complex dynamics and physics associated with multiphase phenomena, particularly in the context of phase change simulations. Unlike many existing datasets that predominantly utilize a single type of boundary condition, BubbleML incorporates a combination of Dirichlet and Neumann boundary conditions [16]. This inclusion enables researchers to explore and model scenarios where multiple boundary conditions coexist, enhancing the realism and applicability of the dataset. Moreover, the presence of "jump" conditions along the liquid-vapor interface adds an additional layer of complexity. These conditions arise due to surface tension effects and require careful modeling to accurately capture the interface behavior [31, 32]. By incorporating such challenges, BubbleML provides a realistic and demanding testbed for ML models.

**Optical Flow Datasets.** Optical flow estimation, a classical ill-posed problem [33] in image processing, has witnessed a shift from traditional methods to data-driven deep learning approaches. Middlebury [34] is a dataset with dense ground truth for small displacements, while KITTI2015 [35] provides sparse ground truth for large displacements in real-world scenes. MPI-Sintel [36] offers synthetic data with very large displacements, up to 400 pixels per frame. However, these datasets are relatively small for training deep neural networks. FlyingChairs [37], a large synthetic dataset, contains around 22,000 image pairs generated by applying affine transformations to rendered chairs on random backgrounds. FlyingThings3D [38] is another large synthetic dataset with approximately 25,000 stereo frames of 3D objects on different backgrounds.

While these datasets have been instrumental in advancing data-driven optical flow methods, they primarily focus on rigid object motion in visual scenes and do not address the specific challenges posed by multiphase simulations. Efforts have been made to capture non-rigid motion in nature, such as piece-wise rigid motions seen in animals [39]. In boiling, the non-rigid dynamics of bubbles and the motion of liquid-vapor interfaces play a crucial role in the distribution and transfer of thermal energy. The BubbleML dataset provides a unique opportunity to explore and develop optical flow algorithms tailored to phase change dynamics. Unlike existing datasets, it offers a diverse range of bubble behaviors, including merging, growing, splitting, and complex interactions (see Figure 1). The ability to accurately predict and forecast bubble dynamics has practical implications in various fields.

## 3 BubbleML: A Multiphase Multiphysics Dataset for ML

We first introduce the preliminary concepts underlying the SciML learning problem and give insights into the simulation and PDEs in this domain. Then, we present an overview of the dataset pipeline.

### 3.1 Preliminaries

A common application for SciML is approximating the solution of *boundary value problems* (BVPs). BVPs are widely used to model various physical phenomena, including fluid dynamics, heat transfer, electromagnetics, and quantum mechanics [16, 40, 41, 42]. BVPs take the form: $L[u(x)] = f(x), x \in \Omega$ and $B[u(x)] = g(x), x \in \partial\Omega$. The goal is to determine the vector-valued solution function, $u$. $x$ is a point in the domain $\Omega$ and may include a temporal component. The boundary of the domain is denoted as $\partial\Omega$. The governing equation is described by the PDE operator $L$, and the forcing function is denoted as $f$. The *boundary condition* (BC) is given by the boundary operator $B$ and the boundary function $g$. $B[u] = g$ ensures the existence and uniqueness of the solution.

There are three common types of BCs: periodic, Dirichlet, and Neumann. Periodic BCs enforce the equality of the solution at distinct points in the domain: $u(x_1) = u(x_2)$. Dirichlet BCs specify the values of the solution on the boundary: $u(x) = g(x)$. Neumann BCs enforce constraints on the derivatives of the solution: $\partial_n u(x) = g(x)$ [16]. As seen in Figure 1, BubbleML combines both Dirichlet (no-slip walls, heater, inflow) and Neumann (outflow) boundaries, which impose constraints on flow and temperature dynamics. Additionally, the "jump conditions" that govern the transitions between the liquid and vapor phases use Dirichlet and Neumann boundaries [31].

### 3.2 Overview of PDEs and Flash-X Simulation

A comprehensive description of the simulations is well beyond the scope of this paper and can be found in [31, 32]. We provide a concise description here as knowledge of the PDEs is important when training physics-informed models.

The liquid ($l$) and vapor ($v$) phases of a boiling simulation are characterized by differences in fluid and thermal properties: density, $\rho$; viscosity, $\mu$; thermal diffusivity, $\alpha$; and thermal conductivity $k$. The phases are tracked using a level-set function, $\phi$, which is positive inside the vapor and negative in the liquid. $\phi = 0$ provides implicit representation of the liquid-vapor interface, $\Gamma$ (see Figure 1). The transport equations are non-dimensionalized and scaled using the values in liquid and are given as,

$$\frac{\partial \vec{u}}{\partial t} + \vec{u} \cdot \nabla \vec{u} = -\frac{1}{\rho'}\nabla P + \nabla \cdot \left[\frac{\mu'}{\rho'}\frac{1}{\text{Re}}\nabla \vec{u}\right] + \frac{\vec{g}}{\text{Fr}^2} + \vec{S}_u^\Gamma + S_P^\Gamma \tag{1a}$$

$$\frac{\partial T}{\partial t} + \vec{u} \cdot \nabla T = \nabla \cdot \left[\frac{\alpha'}{\text{Re Pr}}\nabla T\right] + S_T^\Gamma \tag{1b}$$

where, $\vec{u}$, is the velocity, $P$ is the pressure, and $T$ is the temperature everywhere in the domain. The Reynolds number (Re), Froude number (Fr), and Prandtl Number (Pr) are constants set for each simulation. Scaled fluid properties like $\rho'$ represent the local value of the phase scaled by the corresponding value in liquid. Therefore, $\rho'$ is 1 in liquid phase, and $\rho_v/\rho_l$ for vapor phase. The effect of surface tension is modeled using Weber number (We) and incorporated by a sharp pressure jump, $S_P^\Gamma$, at the liquid-vapor interface, $\Gamma$. The effects of evaporation and saturation conditions on velocity and temperature, $\vec{S}_u^\Gamma$, and $S_T^\Gamma$, are modeled using a ghost fluid method [31]. For a more detailed discussion of non-dimensional parameters and values, we refer the reader to Appendix D.

The continuity equation is given by, $\nabla \cdot \vec{u} = -\dot{m}\nabla(\rho')^{-1} \cdot \vec{n}$, where the mass transfer $\dot{m}$ is computed using local temperature gradients in liquid and vapor phase, $\dot{m} = \text{St}(\text{Re Pr})^{-1}\left[\nabla T_l \cdot \vec{n}^\Gamma - k'\nabla T_v \cdot \vec{n}^\Gamma\right]$ where, $\vec{n}^\Gamma$ is the surface normal vector to the liquid-vapor interface. The Stefan number St, is another constant defined for the simulation and depends on the temperature scaling given by $\Delta T = T_{wall} - T_{bulk}$, and latent heat of evaporation, $h_{lv}$. Simulation data is scaled to dimensional values using the characteristic length $l_0$, velocity $u_0$, and temperature $(T - T_{bulk})/\Delta T$ scale. Temporal integration is implemented using a fractional step predictor-corrector formulation to enforce incompressible flow constraints. The solver has been extensively validated and demonstrates an overall second-order accuracy in space [31, 32].

In thermal science, *heat flux* measured as the integral of the temperature gradient across the heater surface ($\bar{q} = \partial T/\partial y$) serves as a vital indicator of boiling efficiency. It reflects the contribution of multiple sub-processes, such as conduction, convection, microlayer evaporation, and bubble-induced turbulence. Identifying and managing the impact of each sub-process to enhance $\bar{q}$ is an open challenge [4, 43]. *Critical heat flux* (CHF) signifies peak heat flux before a sharp drop in efficiency occurs due to the formation of a vapor barrier (see Figure 5b). It is arguably the most important design and safety parameter for any heat flux controlled boiling application [44]. Accurate heat flux modeling and prediction of boiling crisis are paramount for the reliability of heat transfer systems [45, 46, 47].

The simulations in this study are implemented within the Flash-X framework [21, 31], and a dedicated environment is provided for running new simulations [5]. The repository contains example configuration files for various multiphase simulations, including those used in this dataset. To ensure reproducibility, a lab notework has been designed that organizes each study using configuration files for data curation. The lab notebook and Flash-X source code are open-source to allow for community development and contribution, enabling the creation of new datasets beyond the scope of this paper. The simulation archives store HDF5 output files and bash scripts that document software environment and repository tags for reproducibility. The lab notebook also provides an option to package Flash-X simulations as standalone Docker/Singularity containers, which can be deployed on cloud and supercomputing platforms without the need for installing third-party software dependencies. The latter is ongoing work towards software sustainability [48].

### 3.3 Dataset Overview

The study encompasses two types of boiling – pool boiling and flow boiling. Pool boiling represents fluid confined in a tank above a heater, resembling scenarios like cooling nuclear waste. The BCs for pool boiling include walls on the left and right, an outlet at the top, and a heater at the bottom. In contrast, flow boiling models water flowing through a channel with a heater, simulating liquid

---

[5] `https://github.com/Lab-Notebooks/Outflow-Forcing-BubbleML`

cooling of data center GPUs. There is an inlet BC modeling flow into the system and an outlet. The fluid used for the simulations is FC-72 (perfluorohexane), an electrically insulating and stable fluorocarbon-based fluid commonly used for cooling applications in electronics operating at low temperatures (ranging from $50°C$ to $100°C$). To explore various phenomena, different parameters such as heater temperature, liquid temperature, inlet velocity, and gravity scale are adjusted in each simulation. Table 1 presents a summary of the dataset. Appendix E provides detailed illustrations of the boundary conditions and descriptions of each simulation for reference.

Table 1: Summary of BubbleML datasets and their parameters. $\Delta t$ is the temporal resolution in non-dimensional time ($\Delta t = 1 = 0.008$ seconds). For rationale behind the parameter choices, refer to appendix A.3. PB: pool boiling. FB: flow boiling.

| Dim | Type - Physics | Sims | Domain $(mm^d)$ | Resolution Spatial | $\Delta t$ | Timesteps | Size (GB) |
|---|---|---|---|---|---|---|---|
| 2D | PB - Single Bubble | 1 | $4.2 \times 6.3$ | $192 \times 288$ | 0.5 | 500 | 0.5 |
| 2D | PB - Saturated | 13 | $11.2 \times 11.2$ | $512 \times 512$ | 1 | 200 | 24.2 |
| 2D | PB - Subcooled | 10 | $8.4 \times 8.4$ | $384 \times 384$ | 1 | 200 | 10.3 |
| 2D | PB - Gravity | 9 | $11.2 \times 11.2$ | $512 \times 512$ | 1 | 200 | 16.5 |
| 2D | FB - Inlet Velocity | 7 | $29.4 \times 3.5$ | $1344 \times 160$ | 1 | 200 | 10.7 |
| 2D | FB - Gravity | 6 | $35 \times 3.5$ | $1600 \times 160$ | 1 | 200 | 10.9 |
| 2D | PB - Subcooled$_{0.1}$ | 15 | $8.4 \times 8.4$ | $384 \times 384$ | 0.1 | 2000 | 155.1 |
| 2D | PB - Gravity$_{0.1}$ | 9 | $11.2 \times 11.2$ | $512 \times 512$ | 0.1 | 2000 | 163.8 |
| 2D | FB - Gravity$_{0.1}$ | 6 | $35 \times 3.5$ | $1600 \times 160$ | 0.1 | 2000 | 108.6 |
| 3D | PB - Earth Gravity | 1 | $8.75^3$ | $400^3$ | 1 | 57 | 122.2 |
| 3D | PB - ISS Gravity | 1 | $8.75^3$ | $400^3$ | 1 | 29 | 62.6 |
| 3D | FB - Earth Gravity | 1 | $35 \times 3.5^2$ | $1600 \times 160^2$ | 1 | 55 | 93.9 |

BubbleML stores simulation output in HDF5 files. Each HDF5 file corresponds to the state of a simulation at a specific instant in time and can be directly loaded into popular tensor types (e.g., PyTorch tensors or NumPy arrays) using BoxKit. BoxKit is a custom Python API designed for efficient management and scalability of block-structured simulation datasets [49, 50]. It leverages multiprocessing and cache optimization techniques to improve the read/write efficiency of data between disk and memory. Figure 2 shows an example of a boiling dataset and the corresponding workflow for enabling downstream tasks like scientific machine learning and optical flow. By operating on simulation data in manageable chunks that fit into memory, BoxKit significantly improves computational performance, particularly when handling large datasets.

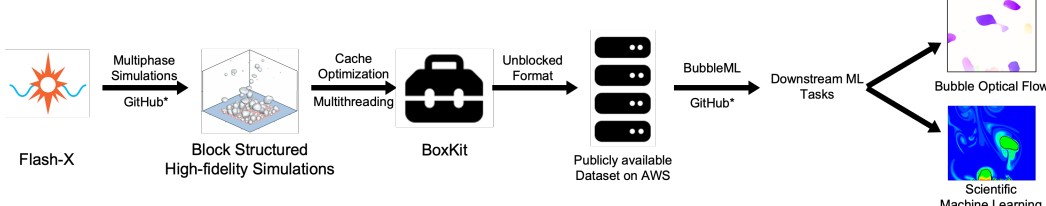

Figure 2: **Dataset Curation and Workflow.** Flash-X multiphase simulations are executed and converted into unblocked HDF5 formats using the BoxKit library. The resulting dataset is publicly available [3], enabling downstream tasks like scientific machine learning and optical flow.

Each simulation within the BubbleML dataset tracks the velocities in the $x$ and $y$ directions, temperature, and a signed distance function (SDF), $\phi$, which denotes the distance from the bubble interface. The SDF can be used to get a mask of the bubble interfaces or determine if a point is in the liquid or vapor phase. These variables are stored in HDF5 datasets. For instance, the temperature is stored as a tensor with a shape $t \times x \times y \times z$ ($t \times x \times y$ in 2D), which allows indexing with $xyz$-spatial coordinates or time. The HDF5 files also include any constants or runtime parameters input to the simulation. Some of these parameters, such as thermal conductivity or Reynolds number, are constants used in the PDEs that govern the system. Including these variables and parameters in the dataset enables comprehensive analysis and modeling of the boiling phenomena.

Table 2: Results of pre-trained and fine-tuned RAFT and GMFlow models on optical flow data (B) generated from `PB-Saturated` dataset. A 80:20 split results in 2000 training and 500 validation image pairs. The pre-trained models include `C` trained on FlyingChairs dataset, `C+T` trained further on FlyingThings3D, and `C+T+S` further fine-tuned for the Sintel test benchmark. `Model+B` represents models fine-tuned on the BubbleML optical flow dataset.

| Model | Method | Chairs (Val) | Things (Val) | Sintel (Train) | | KITTI (Train) | | Boiling (Test) |
|-------|--------|--------------|--------------|----------------|-------|---------------|--------|----------------|
| | | | Clean | Clean | Final | F1-EPE | F1-all | |
| C | RAFT | 0.82 | 9.03 | 2.19 | 4.49 | 9.83 | 37.57 | 4.20 |
| | GMFlow | 0.92 | 10.23 | 3.22 | 4.43 | 17.82 | 56.14 | 4.73 |
| C+B | RAFT | 0.91 | 11.22 | 2.55 | 5.16 | 13.7 | 44.44 | **2.33** |
| | GMFlow | 1.31 | 11.99 | 3.78 | 5.12 | 21.91 | 63.04 | **2.36** |
| C+T | RAFT | 1.15 | 4.39 | 1.40 | 2.71 | 5.02 | 17.46 | 4.72 |
| | GMFlow | 1.26 | 3.48 | 1.50 | 2.96 | 11.60 | 35.62 | 7.98 |
| C+T+B | RAFT | 1.28 | 7.69 | 1.69 | 2.95 | 9.96 | 23.61 | 2.38 |
| | GMFlow | 1.39 | 3.88 | 1.61 | 2.91 | 14.49 | 43.09 | 2.51 |
| C+T+S | RAFT | 1.21 | 4.69 | 0.77 | 1.22 | 1.54 | 5.64 | 8.39 |
| | GMFlow | 1.53 | 4.09 | 0.95 | 1.28 | 3.04 | 13.61 | 14.65 |
| C+T+S+B | RAFT | 1.37 | 6.59 | 0.89 | 1.60 | 1.83 | 6.44 | 2.34 |
| | GMFlow | 1.65 | 4.49 | 1.07 | 1.45 | 4.06 | 18.99 | 2.56 |

BubbleML follows the FAIR data principles [51] as outlined in Appendix A.2. It is essential to validate simulations against experimental observations due to inherent approximations in numerical solvers and simplified models of real-world phenomena. Appendix A.4 provides a detailed validation of the BubbleML dataset.

# 4 Benchmarks of BubbleML: Optical Flow and SciML

## 4.1 Optical Flow

**Generation of Optical Flow Dataset.** Optical flow computes the velocity field of an image based on the relative movement of objects between consecutive frames. This method holds significant implications for downstream tasks, such as extracting side-view boiling statistics and applying SciML to real-world experimental data. Although many datasets capturing spatiotemporal dynamical systems can be repurposed to create optical flow datasets, the inherent non-rigidity of bubbles introduces unique physical phenomena that are not present in other datasets. For instance, consider the scenario where a bubble detaches from a heater surface: the bottom region of the bubble exhibits significantly higher velocity compared to the top region, resulting in a velocity gradient that forces the bubble into a spherical shape. At higher heater temperatures, deformation and detachment processes might occur more frequently, leading to different flow patterns and bubble behaviors illustrated in Figure 1.

We create an optical flow dataset from BubbleML to learn bubble dynamics. Liquid and vapor phases are distinguished using $\phi$. In the generated image sequences, bubble trajectories are tracked across consecutive timesteps. Note that the optical flow dataset only includes bubble velocities at each timestep. Excluding liquid velocities focuses the learning task on capturing observable objects (bubbles). The bubble velocities in non-dimensional units are converted to pixels per frame units (see Appendix B.1) before being written to the widely used Middlebury [34] flow format, resulting in a sequence of images and flow files that resemble the Sintel dataset [36]. For training and validation of optical flow models, PyTorch dataloaders are provided for the generated dataset 3. This allows for easy integration and fine-tuning of existing optical flow models using the BubbleML dataset.

**Learning Bubble Dynamics.** We evaluate and fine-tune two state-of-the-art optical flow models, RAFT [52] and GMFlow [53], using the BubbleML optical flow dataset (B). We consider three different pre-trained models for each method: the first model is trained exclusively on FlyingChairs (C), the second is trained on FlyingChairs and FlyingThings3D (C+T), and the third model is fine-tuned for the Sintel Benchmark (C+T+S). To assess the performance of the trained models, we measure the end-point error. Table 2 summarizes the results for one dataset. Refer to Appendix B for results on the other datasets.

Initially, the pre-trained models exhibit subpar performance on the BubbleML data. To address this, each model is fine-tuned for 3-4 epochs with a low learning rate of $10^{-6}$. After fine-tuning, we observe a significant improvement in predictions for the test data (see Figures 7 and 8 in Appendix B). While all fine-tuned models tend to converge to similar levels of accuracy for pool boiling datasets, fine-tuning the pre-trained FlyingChairs models (C) on the BubbleML (B) dataset gives the best results. This could be attributed to the similar nature of the datasets consisting of 2D objects in motion. In the case of flow boiling, the best results are achieved by fine-tuning models initially trained for the Sintel benchmark (C+T+S). Flow boiling images have an extremely high aspect ratio (8:1), which is similar to the Sintel (3:1) and the KITTI (4:1) datasets. Note that although training the models on the boiling dataset for more epochs improves performance on our specific task, it adversely affects the models' generalization capabilities, leading to increased errors on the other datasets.

**Open problems.** Error analysis (see Appendix B.3) highlights the shortcomings of state-of-the-art optical flow models in accurately capturing the turbulent dynamics of bubbles. Although fine-tuning improves the overall performance, the high errors at the bubble boundaries remain an ongoing challenge. This underscores the need for novel optical flow models that incorporate physical insights to accurately model the complex and chaotic behavior of boiling. BubbleML bridges the gap for physics-informed optical flow datasets.

## 4.2 Scientific Machine Learning

**SciML Prelimaries.** SciML baseline experiments use *neural PDE solvers* to learn temperature and flow dynamics. We focus on two classes of neural PDE solvers: (a) Image-to-image models, widely used in computer vision tasks, such as image segmentation [54]. These may not always be suitable as PDE solvers, since they are limited to fixed resolution, but they remain competitive in many baselines [55, 28]. (b) Neural operators are neural networks that learn a mapping between infinite-dimensional function spaces. As they map functions to functions, neural operators are discretization invariant and can be used on a higher resolution than they were trained [15]. The seminal neural operator is the Fourier Neural Operator (FNO) [13]. Refer to Appendix C for further details.

For both classes of models, we employ the auto-regressive formulation of a forward propagator, denoted as $\mathcal{F}$. For timesteps $\{t_1, \ldots, t_{max}\}$ discretized such that $t_{k+1} - t_k = \Delta t$, the forward propagator $\mathcal{F}$ maps the solution function $u$ at $k$ consecutive time steps $\{t_{m-k}, \ldots, t_{m-1}\}$ to the solution at time $t_m$. For brevity, we use $u([t_{m-k}, t_{m-1}]) = \{u(t_{m-k}), \ldots, u(t_{m-1})\}$. The operator $\mathcal{F}$ can be approximated using a neural network $\mathcal{F}_\theta$ parameterized by $\theta$. This network is trained using a dataset of $N$ ground truth solutions $D = \{u^{(n)}([0, t_{max}]) : n = 1 \ldots N\}$. By applying a standard gradient descent algorithm, we find parameters $\hat{\theta}$ minimizing some loss function of the predictions $\mathcal{F}_{\hat{\theta}}\{u^{(n)}([t_{m-k}, t_{m-1}])\}$ and the ground truth solutions $u^{(n)}(t_m)$. Thus, given solutions for $k$ initial timesteps of an unseen function $u$, we can obtain an approximation $\mathcal{F}_{\hat{\theta}}\{u([0, t_{k-1}])\} \approx u(t_k = t_{k-1} + \Delta t)$. Using this approximation for $t_k$, we can step forward to get $\mathcal{F}_{\hat{\theta}}\{u([t_1, t_k])\} \approx u(t_{k+1})$. This process is called *rollout* and is repeated until reaching $t_{max}$. While in principle, rollout can be done for arbitrary time, the quality of approximation worsens with each step [28, 56]. We implement several strategies that attempt to mitigate this deterioration [57, 58]. However, achieving a long and stable rollout is still an open problem.

**Baseline Implementations.** We implement several baseline image-to-image models—including UNet_bench and UNet_mod—and neural operators—including FNO, UNO, F-FNO, G-FNO, and T-FNO. Detailed descriptions and comparisons of the models are included in Appendix C.1.

**Training Strategies.** Detailed descriptions for each of the training strategies we used are listed in Appendix C.2. We implement teacher-forcing training [59], temporal bundling, and the pushforward trick [57]. Models trained with the pushforward trick are prefixed with "P-". A discussion of hyperparameter settings can be found in Appendix C.3.

**Metrics.** We draw inspiration from PDEBench and adopt a large set of metrics that include the Root Mean Squared Error (RMSE), Max Squared Error, Relative Error, Boundary RMSE (BRMSE), and low/mid/high Fourier errors [17]. These metrics provide a comprehensive view of the physical dynamics, which may be missed when only using a global loss metric. For instance, when predicting temperature, we find that the max error can often be very high due to the presence of sharp transitions between hot vapor and cool liquid. Even a one-pixel misalignment in the model's prediction can cause the reported temperature to be the opposite extreme. Metrics that report a global average

(i.e., RMSE) could mask these errors because they get damped by the average. We incorporate an additional *physics* metric: the RMSE along bubble interfaces (IRMSE). Accuracy along both the domain and immersed boundaries is of significant importance. Boundary conditions determine if the solution to a PDE exists and is unique. In the case of the multiphysics BubbleML dataset, accurate modeling of the system requires satisfying the conditions at the liquid-vapor interfaces accurately.

**Learning Temperature Dynamics.** One application of SciML using the BubbleML dataset is to learn the dynamics of temperature propagation within a system. In this context, the system's velocities serve as a sourcing function, influencing the temperature distribution. Notably, UNet-based models perform best across all datasets (see Figure 3b-d). For a complete listing of error metrics for each model and dataset pairing, refer to Appendix C.4. UNet models may have some advantage in predicting the interfaces and boundaries (IRMSE and BRMSE) because they naturally act as edge-detectors. The temperature also propagates smoothly, so it is likely unnecessary to use global filters like the FNO variants. In contrast, FNO models rely on fast Fourier transforms and weight multiplication in the Fourier space, which, while capable of handling global and local information simultaneously, might not be as effective at capturing local, non-smooth features. Several recent studies report similar observations about auto-regressive UNet and FNO variants [28, 55, 56].

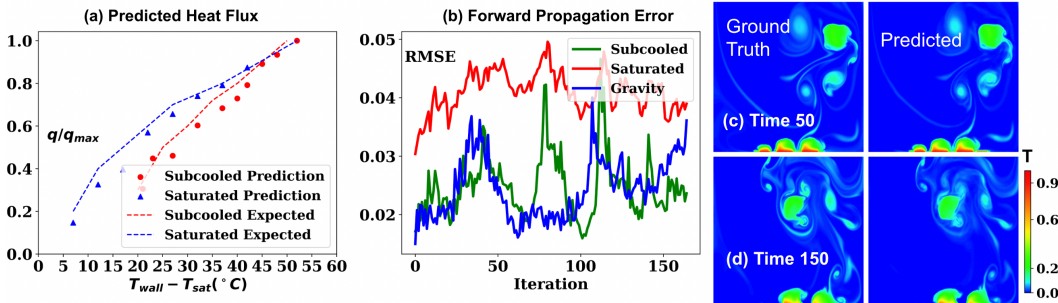

Figure 3: **Temperature and Heat Flux Prediction.** (a) Cross-validated heat flux $q/q_{max}$ estimates for subcooled and saturated boiling. (b), (c), and (d) show results for the fully trained forward propagator. In (b), accuracy degradation is minimal, with spikes occurring during timesteps of violent turbulence caused by rapid bubble detachment from the heater surface. (c) and (d) compare frames from the Flash-X simulation and predictions by the forward propagator for subcooled boiling.

The trained model can be a valuable tool to get fast estimates of heat flux, discussed in section 3.2. Heat flux is influenced by steep temperature gradients and dynamic temporal changes which presents a challenging problem. To further validate our models, we perform cross-validation to predict the heat flux trends observed in Figure 5. For each heat flux prediction, we holdout a simulation and train a forward propagator on the remaining simulations within the dataset. Even with partial training–50 epochs for subcooled boiling models and 100 epochs for saturated boiling models–we achieve compelling results. The heat flux predictions by UNet$_{bench}$ remarkably track the expected trend, as seen in Figure 3a.

**Learning Fluid Dynamics.** As an additional benchmark, we use the BubbleML dataset to train models to approximate both velocity *and* temperature dynamics. A challenging problem! Detailed results are in Appendix C.5. These follow similar training settings to the temperature-only models. Strikingly, we observe nearly the opposite results to predicting only temperature: the UNet$_{bench}$ model struggles when predicting both velocity and temperature fields jointly, while the UNet$_{mod}$ and the FNO variants perform comparatively better. All the models have difficulty capturing the trails of condensation that form in the temperature field. The vapor trails form but dissipate more quickly than expected. An example rollout of UNet$_{mod}$, trained using the pushforward trick, is shown in Figure 4. We see that the flow closely aligns with the ground-truth simulation.

**Open Problems.** We reiterate several open problems in SciML that BubbleML offers an avenue to explore. The first is the creation of a new class of *models that can learn multiple interrelated physics*. We find that while UNet architectures work well at predicting temperature and FNO variants work well at predicting velocity, neither excels at joint prediction of temperature and velocity. The CNN-based UNet architectures slightly outperform FNO and its variants when predicting temperature, potentially due to CNN's capacity to naturally act as edge-detectors, and thus handle non-smooth

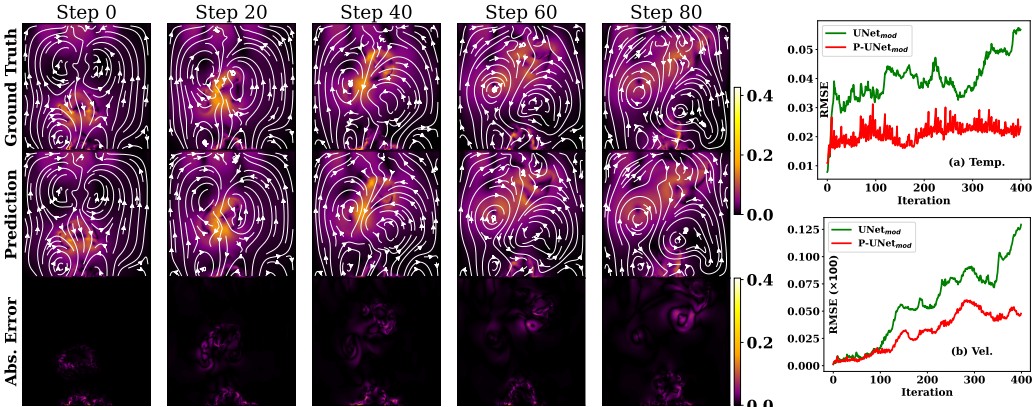

Figure 4: **Velocity and Temperature Rollout.** The left figure shows the first 80 timesteps of P-UNet$_{mod}$'s rollout, where color indicates velocity and streamlines illustrate direction of flow. Both the flow magnitude and direction align exceptionally well with the ground truth. On the right, (a) and (b) show the rollout errors for temperature and velocity predictions. The prefix "P-" denotes that the model is trained with the pushforward trick [57]. Notably, UNet$_{mod}$ starts with slightly better initial accuracy, but it degrades more quickly than the model trained with the pushforward trick. P-UNet$_{mod}$ behaves more stably during rollout.

interfaces more easily. On the other hand, FNO variants perform quite well at predicting velocities but still do not perform particularly well at temperature estimation, especially in capturing condensation trails. This is related to the second problem: *developing neural operators that can handle non-smooth and irregularly shaped interfaces*. FNO variants seem to encounter difficulties in modeling temperature fields, which have sharp jumps along bubble interfaces where the temperature transitions from cool liquid to hot vapor. Conversely, the velocity field appears relatively smooth and thus may be composed of lower frequencies better captured by FNO variants. However, these models still miss sharp and sudden changes in velocity along bubble interfaces that are important for accurately modeling long-range dynamics. The third problem is improving *stability during long rollouts*. This is explored within the context of other datasets [28, 56], but it is particularly relevant for BubbleML. For instance, in subcooled boiling, after bubbles depart from the surface, they undergo condensation and generate vortices that gradually dissipate as they move upstream. To model these extended temporal processes accurately, autoregressive models must be stable across long rollouts. However, we observe that models experience instability, leading them to slowly diverge from the ground truth. The BubbleML dataset presents an opportunity to study these challenges in SciML.

## 5 Conclusions and Limitations

This paper introduces BubbleML, which addresses a critical gap in ML research for multiphase multiphysics systems. By employing physics-driven simulations, the dataset provides precise ground truth information for a number of boiling scenarios, encompassing a wide range of parameters. BubbleML is validated against experimental observations and trends, establishing its reliability in multiphysics phase change research. The two BubbleML benchmarks demonstrate applications in improving the accuracy of optical flow estimation and SciML modeling encountered in multiphase systems. Importantly, BubbleML extends its impact beyond its immediate applications. It resonates with broader challenges in SciML, serving as a foundational platform to study several open problems.

**Limitations.** Combining datasets might pose challenges due to their varying sizes. Because the resolution scales proportionally with the domain size, the constant relative spacing between grid cells allows the UNet model to be effectively trained on the merged boiling dataset. However, this approach does not extend to FNO, requiring domain decomposition methods [60] or downscaling strategies [61] to accommodate variable domain sizes. Note that the dataset is also exclusively composed of simulations due to the unavailability of experimental data with velocity, pressure, and temperature fields. Future work will involve collaboration with experimentalists to augment the dataset.

## Acknowledgments and Disclosure of Funding

This work was partially supported by the National Science Foundation (NSF) under the award number 1750549, the Office of Naval Research (ONR) under grant number N00014-22-1-2063 (supervised by program manager Dr. Mark Spector), the Exascale Computing Project (ECP), a collaborative effort of the US Department of Energy (DOE) Office of Science and the National Nuclear Security Administration (NNSA) under grant number 17-SC-20-SC, and Laboratory Directed Research and Development (LDRD) program at Argonne National Lab under contract number DE-AC02-06CH11357. We thank Kamyar Azizzadenesheli, Colin Wright, and Nikola Kovachki for their suggestions in training FNO. We gratefully acknowledge the GPU computing resources provided on HPC3, a high-performance computing cluster operated by the Research Cyberinfrastructure Center at the University of California, Irvine.

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

# A Additional BubbleML details

## A.1 Dataset URLs and Links

**Dataset:** Links to download all the BubbleML datasets in Table 1 are available on the GitHub homepage. The dataset homepage is hosted at https://hpcforge.github.io/BubbleML/. All future versions of the dataset with the new links will be uploaded here.

**Code:** Code for training and evaluation of all the benchmark models are available within the same GitHub repository.

**Model Weights:** Weights for all the benchmark models are available in the model zoo. All the relevant benchmark results can be accessed on the same page.

**DOI:** The BubbleML dataset has a DOI from Zenodo: https://doi.org/10.5281/zenodo.8039786.

**Documentation:** We provide detailed descriptions for the fields tracked in the simulation data: https://github.com/HPCForge/BubbleML/blob/main/bubbleml_data/DOCS.md. The documentation discusses the layout of the data, important metadata, and some potential pitfalls.

**Tutorials:** We provide example Jupyter notebook to enable reproducibility of the benchmarks and findings in this study: https://github.com/HPCForge/BubbleML/tree/main/examples. These tutorials cover accessing simulation data, dataset schema, querying simulation parameters, and training a Fourier Neural Operator—discussed in Section C.1—using PyTorch.

## A.2 Maintenance and Long Term Preservation

The authors of BubbleML are committed to maintaining and preserving this dataset. The authors will likely make extensions as we use BubbleML for future research. Ongoing maintenance also encompasses tracking and resolving issues identified by the broader community after release. User feedback will be closely monitored via the GitHub issue tracker. All data is hosted on AWS, which guarantees reliable and stable storage. Depending on usage, we may migrate to archival storage for long-term preservation.

**Findable:** All data is stored in an Amazon AWS S3 instance. All present and future data will share a global and persistent DOI https://doi.org/10.5281/zenodo.8039786.

**Accessible:** All data and descriptive metadata can be downloaded from the public links listed on the GitHub homepage. For added convenience, we provide a bash script for users to download the entire dataset at once.

**Interoperable:** All BubbleML data is provided in the form of standard HDF5 files that can be read using many common libraries, such as h5py for Python. Relevant metadata is stored with each simulation.

**Reusable:** BubbleML is released under the Creative Commons Attribution 4.0 International License.

## A.3 Justification of simulation parameters

We discuss the rationale guiding the selection of simulation parameters in Table 1, addressing both the quantity of simulations and their resolutions/timesteps.

**Number of simulations:** The studies performed in BubbleML encompass diverse two-phase boiling phenomena. The number of simulations is determined based on the distribution of the variable being studied, keeping other factors constant. For example, in the case of saturated boiling, we choose a wall temperature range from $60°C$ to $120°C$, with uniform intervals of $5°C$, resulting in 13 simulations. This range captures the boiling transition from the bubbly regime to chaotic dynamics as the heat flux approaches criticality. However, when studying the effects of gravity, the scaling factor, $Fr^{-2}$ (Fr is the Froude Number), is chosen from the range $10^{-4}$ to 1 using a logarithmic scale resulting in 9 simulations. This scale is essential to cover the vastly different gravity conditions spanning from the Earth's surface to the International Space Station.

**Domain size:** Exploring phenomena across varying scales and geometries is common practice. Such variations in sizes enable the exploration of a broad spectrum of heat transfer dynamics, bubble

dynamics, and phase change behaviors. The domain and heater sizes are chosen to replicate typical ranges in experiments [62, 63] while also taking into account the computational costs of simulations.

**Spatial resolution:** The domain resolution depends on the grid size of an individual 2D block. A block with spatial dimensions of $0.5 \times 0.5$ (in non-dimensionalized units) is discretized into a grid of size $16 \times 16$. This resolution is determined based on grid sensitivity studies conducted for a single bubble case to ensure high-fidelity simulations [31]. This results in the spatial resolution sizes in Table 1.

**Temporal resolution and timesteps:** The temporal discretization in the majority of BubbleML datasets is set at 1 non-dimensional unit, equivalent to 0.008 seconds for FC-72. Additionally, we include several datasets with a finer discretization of 0.1 non-dimensional time. In both cases, we intentionally chose a discretization that is much larger than what the CFL condition mandates for the Flash-X solver. A significant advantage of neural PDE solvers is their ability to maintain reasonable approximations while taking much larger timesteps than traditional numerical simulations. However, this choice introduces a trade-off between dataset size and ease of use. The datasets with a 1 time unit discretization are potentially more accessible but might present challenges in achieving accurate results. Conversely, the datasets with a 0.1 time unit discretization are less accessible and may require distributed training. Yet, they are likely to achieve more accurate results due to more training data (i.e., more timesteps) and the presence of fine-grained physics.

### A.4  Dataset Validation

**Saturated and subcooled boiling.**  We first validate two distinct boiling phenomena. Saturated boiling refers to the state of a liquid when it reaches its boiling point, known as the saturation temperature, $T_{sat}$. At this temperature, the liquid is in equilibrium with its vapor phase, and bubbles start forming at the heated surface. The liquid is on the verge of vaporization, and any further increase in temperature can lead to the formation of vapor bubbles. This bubble formation is called nucleate boiling; a far more effective way to transfer heat than natural convection on its own. On the other hand, subcooled boiling is a complex process with evaporation and condensation occurring simultaneously. The heat flux imposed on the wall produces a thermal layer around it in which bubbles may nucleate and grow. However, condensation occurs as a bubble migrates into the bulk liquid region with temperature under saturation point, $T_{bulk} < T_{sat}$. Experimental findings [64] indicate that the heat flux increases linearly with the heater temperature, $T_{wall}$ until reaching CHF, marking the transition from nucleate pool boiling to film boiling.

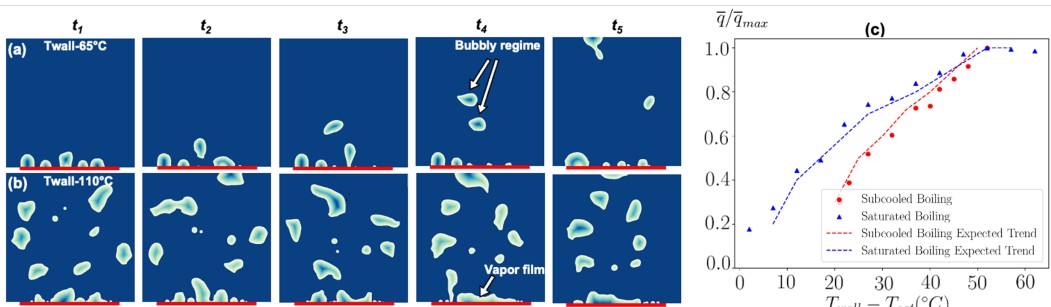

Figure 5: Bubble dynamics for saturated and subcooled pool boiling at different wall temperatures. (a) ONB during saturated boiling is marked by a bubbly flow regime. (b) CHF behavior, indicated by chaotic bubble dynamics and vapor film covering the heater surface. (c) Normalized heat flux, $\overline{q}/\overline{q}_{max}$, plotted against temperature difference between heater and liquid saturation temperature ($T_{sat} = 58°C$) aligns with the boiling curve for FC-72 in [64].

Figure 5 presents the bubble dynamics in two different regimes of saturated pool boiling: onset of nucleate boiling (ONB) and CHF. ONB occurs at low wall superheat and exhibits structured bubbly flow with consistent shape and size of departing bubbles from the heater surface. In contrast, the CHF regime is characterized by chaotic bubble dynamics. Remarkably, the heat flux, $\overline{q}$ at the heater surface for both subcooled and saturated boiling shown in Figure 5c closely match the experimental boiling curve, specifically for Si(100) surface [64]. Beyond CHF, the heat flux reaches a plateau, indicating a stasis marked by the presence of large pockets of vapor cover on the heater surface, as shown in Figure 5b.

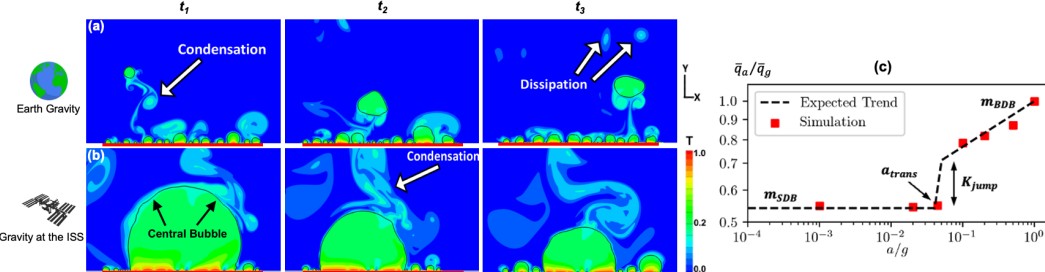

Figure 6: Gravity scaling of subcooled pool boiling: Bubble dynamics (black lines) and temperature distribution (contours) on (a) Earth gravity, $a/g = 1.0$, $t_1 < t_2 < t_3$ and (b) low gravity, $a/g = 0.001$. (c) Wall heat flux vs. gravity for 2D simulations and its comparison to the scaling model in [62].

**Boiling at low gravity.**    Boiling is the most efficient mode of heat transfer on Earth and serves as a cooling mechanism for various thermal applications. However, in low gravity environments such as the International Space Station (ISS), the dynamics of bubble growth, merger, and departure, which significantly impact thermal efficiency, are influenced by the interplay between surface tension and gravity. Quantifying these effects is crucial for developing phase change heat transfer systems in such environments.

Figure 6 provides an overview of simulations conducted to compare against the gravity-based heat flux model proposed in [62, 63]. These simulations cover a range of gravity levels, $Fr = 1 - 100$. The gravity scaling, $a/g = \frac{1}{Fr^2}$, is used to scale values relative to the Earth's gravity ($g$). Gravity separates pool boiling into two distinct regimes: buoyancy dominated boiling (BDB) and surface tension dominated boiling (SDB). The transitional acceleration, $a_{trans}$, which depends on the size of the heater, serves as the boundary between these two regimes.

In BDB, bubbles periodically detach from the heater surface when buoyancy takes over surface tension. Figure 6 illustrates the temporal evolution of temperature and the liquid-vapor interface during bubble departure events for $a/g = 1$. After departing from the surface, the bubbles undergo condensation due to subcooling, and generate vortices that gradually dissipate as they move upstream toward the outflow. This dissipation occurs as the vapor completely condenses into liquid. Decreasing gravitational acceleration results in larger departing bubbles and reduces the wall heat flux, $\bar{q}$. The scaling of $\bar{q}$ with respect to gravity, $\bar{q}_a/\bar{q}_g$, follows the slope $m_{BDB}$. In SDB, the dynamics are dominated by the presence of a central bubble that remains on the heater surface and acts as a vapor sink for smaller satellite bubbles, as depicted in Figure 6 for $a/g = 0.001$. The figure also captures the transient behavior of the central bubble, which fluctuates in size due to the balance between evaporation and condensation leading to different types of vortical structures. The heat flux drops sharply by a value $K_{jump}$, which depends on the size of the central bubble [32], and the slope, $m_{SDB} = 0$. The gravity scaling of heat flux computed from simulations accurately matches the expected trend from the model [62, 63], validating the simulation results.

## B    Optical Flow

### B.1    Dataset

The BubbleML velocity data is in non-dimensional units (refer to appendix D). The non-dimensional velocities are first converted to real-world units by multiplying them with the characteristic velocity of the simulation. The velocities are then multiplied by a scaling factor (number of pixels per unit length in the image) and then divided by the frames per second value. This converts the velocities to pixels per frame units compatible with state-of-the-art optical flow models.

### B.2    Benchmark Models

We consider two state-of-the-art optical flow models, RAFT [52] and GMFlow [53].

1. **RAFT**, Recurrent All-Pairs Field Transforms (RAFT), processes per-pixel features and forms multi-scale 4D correlation volumes between all pixel pairs. The flow field is iteratively updated through a recurrent unit that performs lookups on the correlation volumes. RAFT

begins with a feature encoder that extracts a feature vector for each pixel. This is followed by a correlation layer that produces a 4D correlation volume for all pairs of pixels. Subsequently, this data is pooled to create lower resolution volumes. Finally, a GRU-based update operator utilizes the data from the correlation volumes to iteratively refine the flow field, starting its calculations from zero.

2. **GMFlow** reformulates optical flow as a global matching problem, which identifies the correspondences by directly comparing feature similarities. It extracts dense features from a convolutional backbone network, uses a transformer for feature enhancement, a correlation and softmax layer for global feature matching, and a self-attention layer for flow propagation. It also has a refinement step to exploit higher resolution features, reusing the GMFlow framework for residual flow estimation.

### B.3  Learning Bubble Dynamics Results

**Analysis.** For pool boiling test images, the models trained on FlyingChairs and fine-tuned on Boiling images (C+B) achieve the best performance, as seen from Tables 2, 3, and 4. However, for flow boiling test images, a different trend emerges. As observed in Tables 5 and 6, the lowest errors are achieved by fine-tuning the Sintel benchmark models (C+T+S+B).

Figures 7 and 8 show a visual comparison between the outputs of the best fine-tuned and pre-trained models and the corresponding ground truth optical flows for RAFT and GMFlow respectively. In many instances (highlighted in red), the pre-trained models fail to capture the direction of movement of bubbles which are visibly corrected by fine-tuning. However, as seen in the ground truth optical flows, the bubble boundaries generally have different values than the interior regions. This gradient is responsible for the dynamic changes in bubble shapes between consecutive frames. The end point errors at these bubble-liquid interfaces remain significantly large even after fine-tuning. The bubble-liquid interface is the most important region, where a highly accurate optical flow prediction is necessary to enable downstream tasks such as scientific machine learning on experimental boiling data. This suggests the need to incorporate physics-informed learning rules and architectures into optical flow models to improve the interface end point errors.

Table 3: Results on optical flow data (B) generated from `PB-Subcooled` simulations. It consists of around 1100 training image pairs and around 300 test image pairs. It is to be noted that heater temperatures lower than $90°C$ are not used due to the lack of trackable bubbles in many frames.

| Model | Method | Chairs(Val) | Things(Val) | Sintel(Train) | | KITTI(Train) | | Boiling(Test) |
|---|---|---|---|---|---|---|---|---|
| | | | Clean | Clean | Final | F1-EPE | F1-all | |
| C | RAFT | 0.82 | 9.03 | 2.19 | 4.49 | 9.83 | 37.57 | 2.41 |
| | GMFlow | 0.92 | 10.23 | 3.22 | 4.43 | 17.82 | 56.14 | 1.92 |
| C+B | RAFT | 0.89 | 11.64 | 2.58 | 5.17 | 14.01 | 47.53 | **0.63** |
| | GMFlow | 1.34 | 12.23 | 3.93 | 5.27 | 22.83 | 64.41 | **0.63** |
| C+T | RAFT | 1.15 | 4.39 | 1.40 | 2.71 | 5.02 | 17.46 | 1.91 |
| | GMFlow | 1.26 | 3.48 | 1.50 | 2.96 | 11.60 | 35.62 | 4.76 |
| C+T+B | RAFT | 1.24 | 6.45 | 1.54 | 2.82 | 7.35 | 20.27 | 0.70 |
| | GMFlow | 1.46 | 4.04 | 1.66 | 3.11 | 14.50 | 44.35 | 0.67 |
| C+T+S | RAFT | 1.21 | 4.69 | 0.77 | 1.22 | 1.54 | 5.64 | 6.07 |
| | GMFlow | 1.53 | 4.09 | 0.95 | 1.28 | 3.04 | 13.61 | 9.21 |
| C+T+S+B | RAFT | 1.35 | 6.49 | 0.87 | 1.48 | 1.79 | 6.31 | 0.64 |
| | GMFlow | 1.66 | 4.43 | 1.04 | 1.42 | 3.99 | 18.83 | 0.65 |

**Compute Resources**. All models are fine-tuned using a single NVIDIA V100 GPU. The implementations use the recommended versions of PyTorch and other libraries used in the official repositories of GMFlow and RAFT.

Table 4: Results on optical flow data (B) generated from `PB-Gravity` simulations. It consists of around 1400 training image pairs and around 360 test image pairs.

| Model | Method | Chairs(Val) | Things(Val) | Sintel(Train) | | KITTI(Train) | | Boiling(Test) |
|---|---|---|---|---|---|---|---|---|
| | | | Clean | Clean | Final | F1-EPE | F1-all | |
| C | RAFT | 0.82 | 9.03 | 2.19 | 4.49 | 9.83 | 37.57 | 3.30 |
| | GMFlow | 0.92 | 10.23 | 3.22 | 4.43 | 17.82 | 56.14 | 2.40 |
| C+B | RAFT | 0.95 | 10.98 | 2.78 | 5.76 | 15.52 | 49.88 | **0.90** |
| | GMFlow | 1.42 | 12.57 | 4.06 | 5.41 | 23.61 | 66.38 | **0.90** |
| C+T | RAFT | 1.15 | 4.39 | 1.40 | 2.71 | 5.02 | 17.46 | 2.75 |
| | GMFlow | 1.26 | 3.48 | 1.50 | 2.96 | 11.60 | 35.62 | 3.60 |
| C+T+B | RAFT | 1.31 | 6.65 | 1.64 | 2.93 | 8.39 | 21.36 | 0.96 |
| | GMFlow | 1.41 | 3.95 | 1.64 | 2.97 | 44.07 | 14.41 | 0.97 |
| C+T+S | RAFT | 1.21 | 4.69 | 0.77 | 1.22 | 1.54 | 5.64 | 4.74 |
| | GMFlow | 1.53 | 4.09 | 0.95 | 1.28 | 3.04 | 13.61 | 5.49 |
| C+T+S+B | RAFT | 1.35 | 6.44 | 0.90 | 1.49 | 1.81 | 6.27 | 0.92 |
| | GMFlow | 1.63 | 4.43 | 1.03 | 1.40 | 4.29 | 20.93 | 0.92 |

Table 5: Results on optical flow data (B) generated from `FB-Inlet Velocity` simulations. It consists of around 1200 training image pairs and around 300 test image pairs.

| Model | Method | Chairs(Val) | Things(Val) | Sintel(Train) | | KITTI(Train) | | Boiling(Test) |
|---|---|---|---|---|---|---|---|---|
| | | | Clean | Clean | Final | F1-EPE | F1-all | |
| C | RAFT | 0.82 | 9.03 | 2.19 | 4.49 | 9.83 | 37.57 | 16.01 |
| | GMFlow | 0.92 | 10.23 | 3.22 | 4.43 | 17.82 | 56.14 | 21.44 |
| C+B | RAFT | 1.21 | 14.99 | 3.62 | 6.78 | 22.07 | 60.64 | 10.13 |
| | GMFlow | 1.50 | 13.96 | 4.52 | 5.89 | 23.45 | 65.96 | 7.01 |
| C+T | RAFT | 1.15 | 4.39 | 1.40 | 2.71 | 5.02 | 17.46 | 25.14 |
| | GMFlow | 1.26 | 3.48 | 1.50 | 2.96 | 11.60 | 35.62 | 19.39 |
| C+T+B | RAFT | 1.62 | 9.09 | 2.19 | 3.77 | 13.81 | 32.13 | **9.19** |
| | GMFlow | 1.45 | 4.15 | 1.78 | 3.05 | 15.74 | 48.34 | 7.24 |
| C+T+S | RAFT | 1.21 | 4.69 | 0.77 | 1.22 | 1.54 | 5.64 | 21.23 |
| | GMFlow | 1.53 | 4.09 | 0.95 | 1.28 | 3.04 | 13.61 | 47.88 |
| C+T+S+B | RAFT | 2.18 | 9.32 | 1.54 | 2.90 | 2.57 | 10.93 | 9.68 |
| | GMFlow | 1.72 | 4.80 | 1.18 | 1.65 | 4.69 | 24.56 | **6.88** |

Table 6: Results on optical flow data (B) generated from `FB-Gravity` simulations. It consists of around 1000 training image pairs and around 250 test image pairs.

| Model | Method | Chairs(Val) | Things(Val) | Sintel(Train) | | KITTI(Train) | | Boiling(Test) |
|---|---|---|---|---|---|---|---|---|
| | | | Clean | Clean | Final | F1-EPE | F1-all | |
| C | RAFT | 0.82 | 9.03 | 2.19 | 4.49 | 9.83 | 37.57 | 20.42 |
| | GMFlow | 0.92 | 10.23 | 3.22 | 4.43 | 17.82 | 56.14 | 14.05 |
| C+B | RAFT | 1.14 | 12.65 | 3.30 | 6.13 | 17.94 | 53.76 | 4.45 |
| | GMFlow | 1.30 | 13.02 | 4.05 | 5.41 | 23.17 | 65.33 | 4.24 |
| C+T | RAFT | 1.15 | 4.39 | 1.40 | 2.71 | 5.02 | 17.46 | 21.04 |
| | GMFlow | 1.26 | 3.48 | 1.50 | 2.96 | 11.60 | 35.62 | 13.37 |
| C+T+B | RAFT | 1.41 | 6.85 | 1.97 | 3.89 | 10.96 | 27.82 | 4.24 |
| | GMFlow | 1.37 | 3.89 | 1.71 | 3.05 | 14.29 | 45.03 | 3.94 |
| C+T+S | RAFT | 1.21 | 4.69 | 0.77 | 1.22 | 1.54 | 5.64 | 21.04 |
| | GMFlow | 1.53 | 4.09 | 0.95 | 1.28 | 3.04 | 13.61 | 22.84 |
| C+T+S+B | RAFT | 1.59 | 7.29 | 1.13 | 2.14 | 2.22 | 9.34 | **4.12** |
| | GMFlow | 1.62 | 4.42 | 1.04 | 1.43 | 3.86 | 17.76 | **3.93** |

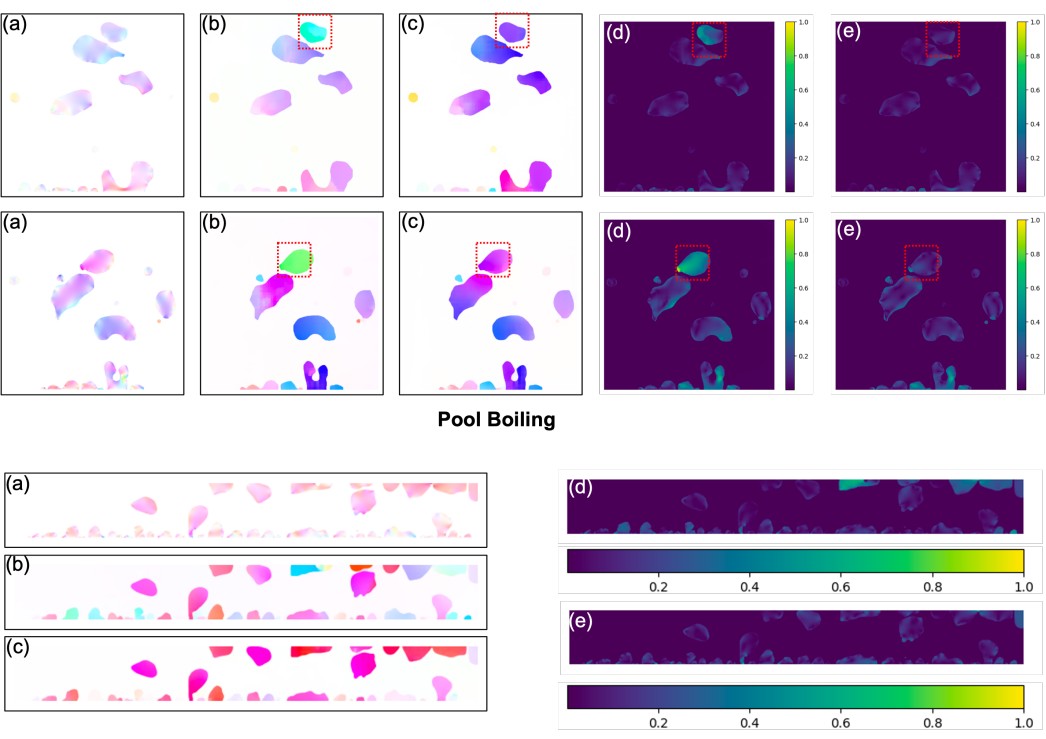

Figure 7: Comparison of the best performing pretrained RAFT models with finetuned RAFT models on Pool Boiling and Flow Boiling images. In each row, (a) is the ground truth optical flow, (b) the optical flow predictions from the pretrained model, and (c) the optical flow predictions from the finetuned model. (d) shows the normalized end-point error at every pixel for the pretrained model output. (e) shows the normalized end-point error at every pixel for the finetuned model output

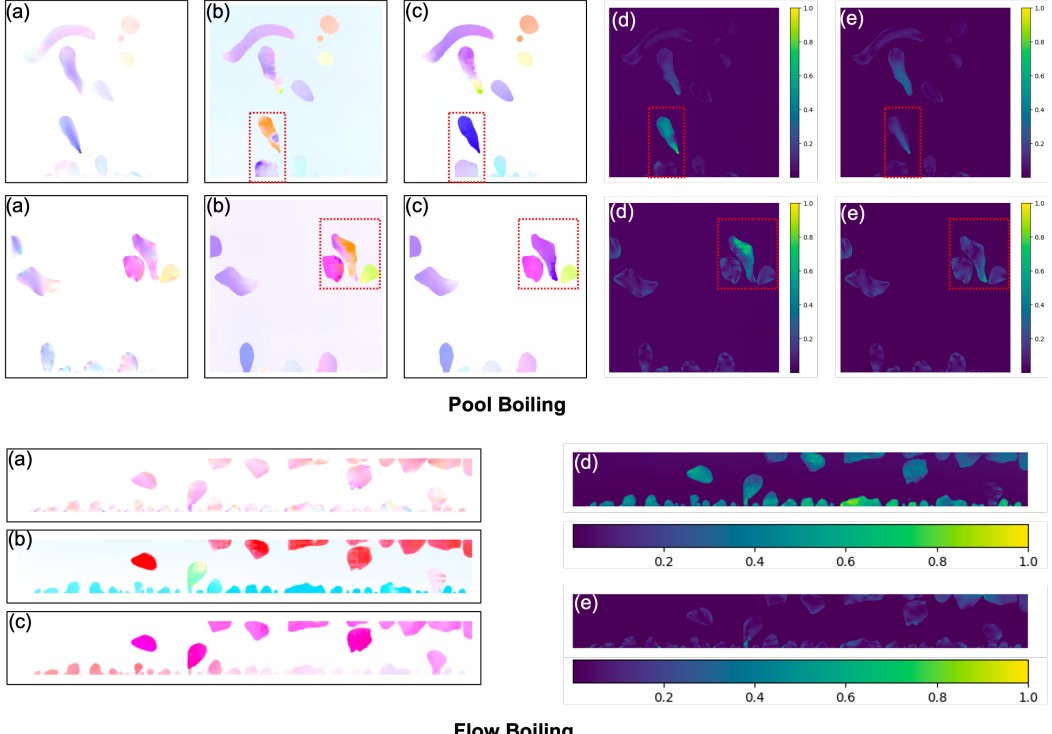

**Pool Boiling**

**Flow Boiling**

Figure 8: Comparison of the best performing pretrained GMFlow models with finetuned GMFlow models on Pool Boiling and Flow Boiling images. In each row, (a) is the ground truth optical flow, (b) the optical flow predictions from the pretrained model, and (c) the optical flow predictions from the finetuned model. (d) shows the normalized end-point error at every pixel for the pretrained model output. (e) shows the normalized end-point error at every pixel for the finetuned model output

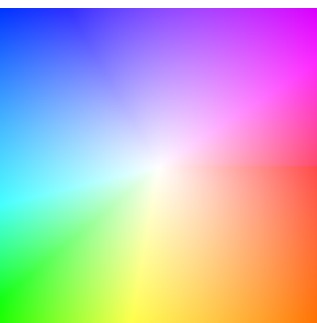

Figure 9: Flow field color coding used in the above images. The displacement of every pixel is the vector from the center of the square to this pixel.

# C  Scientific Machine Learning

## C.1  Benchmark Models

We focus on two classes of neural PDE solvers: Image-to-image models and neural operators. The Image-to-image models are variants of UNet, commonly used in computer vision tasks, such as image segmentation. These models are not always suitable as neural PDE solvers, where training data generated from numerical simulations may require using different resolutions. However, modern versions of UNet are still competitive in many benchmarks [55, 28].

1. **UNet**$_{\text{bench}}$ is a simple variant of the original UNet architecture from 2015, which was modified in PDEBench [17] to use batch normalization. We also replace the tanh activation with GELU [17, 54, 65].

2. **UNet**$_{\text{mod}}$ is a modern variant of UNet that gets impressive as a PDE solver. It has been modified to use wide residual connections and group normalization [55, 66, 67].

Neural operators have found great success as PDE solvers [15]. Typically, given an initial condition $u_0$, a neural operator is a function $\mathcal{M} : [0, t_{max}] \times \mathcal{A} \rightarrow \mathcal{A}$, where $\mathcal{A}$ is an infinite dimensional function space, that satisfies $\mathcal{M}(t, u_0) = u_t$ [13, 15]. I.e., they map some initial condition to any (discretized) time up to $t_{max}$. Training a model in this form demands many valid initial conditions and solved simulations to form a large training set. However, generating such a dataset proves infeasible in the case of BubbleML due to the complexity and computational cost of the simulations. To overcome this obstacle, we adopt the autoregressive approach, even for neural operators, a practice also observed in [28, 56, 57].

1. **FNO** is the Fourier Neural Operator [13, 15]. FNO learns weights in Fourier space to act as a resolution-invariant global convolution. This seminal work has been extended by many other neural operators. One notable downside is the memory requirements: each of the Fourier domain weight matrices consumes $\mathcal{O}(H^2 M^D)$, where $H$ is the hidden size, $M$ is the number of Fourier modes used after truncating high frequencies, and $D$ is the problem dimension. In the case of BubbleML, $D$ is 2 or 3.

2. **UNO** is a U-shaped variant of FNO [68]. This mimics the skip connections of a standard UNet but replaces the convolutional layers with Fourier blocks. We use an 8-layer architecture that mimics the vanilla UNet. The encoder reduces the input size and Fourier modes but increases the number of channels. The decoder increases the input size and Fourier modes while decreasing the number of channels. Since the number of Fourier modes decreases, much of the network only accounts for very low frequencies.

3. **F-FNO** is the Factorized Fourier Neural Operator [58]. As the name suggests, F-FNO factorizes the Fourier transform over problem dimensions. This reduces the number of parameters per Fourier space weight matrix to $\mathcal{O}(H^2 MD)$. This enables using more modes, scaling to deeper models, or learning higher dimensional problems. Further, F-FNO suggests changing the structure of the residual connections, so that they are applied after the non-linearity.

4. **G-FNO** is the Group-Equivariant Fourier Neural Operator [69], which extends group convolutions to the frequency domain. The proposed G-Fourier layers are equivariant to rotation, reflection, and translation. They also preserve the resolution invariance of neural operators. The Fourier space weight matrix has $\mathcal{O}(H^2 M^D S)$ parameters, where $S$ is the number of elements in the "stabilizer" of the group under consideration [69]. We focus on the symmetry group $p4$, which captures $90°$ rotations and translation. It's worth noting that the BubbleML simulations are *not* rotation equivariant due to the global effects of temperature and gravity. However, it is not uncommon for models that bake in some assumption of equivariance to perform well, even in cases where such assumptions are not entirely valid. Despite this incompatibility, we observe that G-FNO can still be successfully applied on BubbleML and achieves performance that matches or surpasses the standard FNO.

5. **T-FNO** is the "Tuckerized" Fourier Neural Operator [70], which replaces FNO's dense weight matrix with a low-rank approximation. Tucker decomposition significantly reduces the number of parameters while maintaining accuracy. Notably, T-FNO has been shown to improve model performance in low-data regimes [70]. This is particularly relevant for

BubbleML, where the complexity of simulations limits the size of the available dataset. The concept of decomposing the weight matrix can potentially be extended to other models, such as G-FNO, to yield comparable results.

## C.2 Training Strategies

We implement several popular training strategies designed to make neural PDE solvers robust to instability during rollout. One source of this instability is *distribution shift*, which occurs when the model takes its prior outputs as input. The errors in the model's predictions accumulate, and the model is likely to diverge from the ground truth during long rollouts.

**Teacher-forcing** is the default training strategy [59]. The input to the model is always the ground truth of the previous timesteps. I.e., to predict the solution $u(t)$, the model input is the ground-truth solutions $u([t - k, t - 1])$.

**Gaussian noise** is applied on iterations that the pushforward trick is unused. Several papers claim that adding noise is a simple method to help the model become more robust to distribution shift during longer rollouts [58, 71, 72]. Similar to F-FNO, when used, we add noise with zero mean and standard deviation of $0.01$. As Gaussian noise is spread uniformly across frequencies, it may also help the model account for high frequencies [28].

**Pushforward trick.** Algorithm 1 shows sample pseudo-code for the pushforward trick [57]. The model is trained by performing multiple inferences for successive timesteps. Similar to using Gaussian noise, this trick helps the model be robust to distribution shifts during rollouts. The advantage is that the errors introduced during the pushforward steps better match the distribution of errors that accumulate during rollouts. Thus, we are essentially training the model to correct for its own distribution shift during rollout. We use a ramp-up when applying the pushforward trick: the percentage of training iterations completed is used as the likelihood that we perform pushforward steps. Initially, there is a $0\%$ chance of pushing forward. After $50\%$ of training iterations, there is a $50\%$ chance of applying the pushforward trick. We found this to be necessary and that applying the pushforward trick on every iteration would fail.

---
**Algorithm 1** Pushforward Trick.

---
**Require:** input is the model's input; $N$ is the number of pushforward steps
 1: **for** $n \in \{0, \ldots, N\}$ **do**
 2:     input $\leftarrow$ model(input)                         ▷ Step through $N$ timesteps
 3: **end for**
 4: input $\leftarrow$ model(input)             ▷ Track gradients for only the final forward pass.

---

**Temporal bundling** is used so that the model outputs the solution for multiple future timesteps with one inference [57]. By having the model output the solution for multiple future timesteps, we can reduce the number of times the model is used. Since some error is introduced by each inference during rollout, temporal bundling should reduce the rate that error is accumulated. This is again beneficial for maintaining accuracy and stability during long rollouts. With a temporal bundle of size $k$, the forward propagator outputs a vector $\mathcal{F}(u([t_{n-k}, t_{n-1}])) = u([t_n, t_{n+k-1}])$.

---
**Algorithm 2** Main Training Loop, with Pushforward Trick.

---
**Require:** data, the set of simulations; $N$, the number of pushforward steps
 1: sample $t$ from $\{t_k, t_{max}\}$
 2: input $\leftarrow$ data($[t - k\Delta t, t - \Delta t]$)               ▷ Sample $k$ timesteps to use as input
 3: **if** $N = 0$ **then**
 4:     input $\leftarrow$ input $+ \mathcal{N}(0, 0.01)$        ▷ Add Gaussian noise when *not* "pushing forward."
 5: **end if**
 6: output = PushForwardTrick(input, $N$)
 7: target $\leftarrow$ data($[t + Nk, t + (N + 1)k - 1]$)     ▷ Grab target for result of $N$ pushforward steps
 8: loss(output, target)

---

Table 7: Generic hyperparameter settings for the temperature experiments.

| Hyperparameter | Value |
|---|---|
| Number of Epochs | 250 |
| Batch Size | 8 |
| Optimizer | AdamW |
| Weight Decay | 0.01 |
| Base LR | 1e-3 (or 5e-4) |
| LR Warmup | Linear, 3% |
| LR Scheduler | Step |
| Step Factor | 0.5 |
| Step Patience | 75 Epochs |
| History Size | 5 |
| Future Size | 5 |

Table 8: Generic hyperparameter settings for the flow experiments.

| Hyperparameter | Value |
|---|---|
| Number of Epochs | 25 |
| Batch Size | 16 |
| Optimizer | AdamW |
| Weight Decay | 1e-4 |
| Base LR | 5e-4 |
| LR Warmup | Linear, 3% |
| LR Scheduler | Cosine Annealing |
| Min LR | 1e-6 |
| History Size | 5 |
| Future Size | 5 |

## C.3 Hyperparameter Settings

We mostly use generic hyperparameter settings that we found work well across all models. Each individual model, however, has its own specific parameters, such as hidden channels, Fourier modes, etc. These parameters are tuned on the subcooled Pool Boiling dataset and reused for the other datasets. The settings for the temperature and flow experiments can be found in Tables 7 and 8. All models are trained using AdamW [73]. For each temperature model, we use 3% of the total iterations for learning rate warmup to reach the base learning rate. For the remaining iterations, we apply a learning rate decay. For UNet$_{mod}$, we reduce the base learning rate to 5e-4 in order to avoid instability we experienced during some training runs. We also perform gradient clipping to clip the gradient $l_2$-norm to be at most one. For data augmentation, we perform horizontal reflections only for pool boiling.

The Fourier models that are resolution invariant are trained at half resolution and evaluated at full resolution. We found that training at the full resolution gave similar results, but took much longer or required distributed training. The Fourier models all take the $xy$-coordinate grid as input [69]. So, the input is $(x, y, T([t - k, t - 1]), v([t - k, t - 1]))$, where $T$ is the temperature map and $v$ is the velocity map. Each Fourier model uses the 64 lowest frequency modes, except UNO, which uses the 32 lowest frequencies in the bottom of the "U" where the resolution is too small to use 64 modes. As the domain is non-periodic, we do Fourier continuation by padding the domain with zeros [13]. FNO is tuned with varying number of hidden channels. We achieve the best results with four layers and 64 hidden channels per layer. For G-FNO, we use the same settings, but halve the number of hidden channels to account for the increase in size of the weight tensor.

## C.4 Learning Temperature Dynamics Results

We show the complete listing of results for predicting temperature dynamics in Tables 9, 10, 11, 12, and 13. As a general trend, we see that UNet$_{bench}$ consistently performs the best. In some metrics, UNet$_{mod}$ closely matches or is better than UNet$_{bench}$, but it tends to have higher errors at bubble interfaces and domain boundaries. The FNO variants also perform well, but slightly worse than

the UNet architectures. The better performance of UNet architecture might stem from the nature of convolutions to function as effective edge detectors. Thus, the UNet variants can handle sharp interfaces between liquid and vapor. Since the Fourier layers implicitly assume periodicity, they tend to act as "smoothers". Qualitatively, they appear to soften the sharp interfaces, resulting in errors along the region surrounding the bubbles. Furthermore, it's important to note that in FNO, frequency truncation primarily preserves the low-frequency modes. While this is well-suited for modeling and capturing general patterns and large-scale dynamics, it may not adequately represent the small-scale structures present around bubble interfaces, the heater surface, and condensation trails. There is existing work that makes similar observations [55, 56]. In our efforts to train FNO for the solution of complex PDEs with high-frequency information, we have experimented with increasing the number of effective Fourier modes to 2/3 of the training data resolution [74]. These results also underscore the importance of regularization in low-data regimes. Among the FNO variants, those that effectively reduce the number of parameters while retaining model expressiveness (T-FNO and F-FNO) outperformed their larger, dense counterparts (FNO, UNO, and G-FNO). Simply reducing the number of parameters in these three models did not lead to substantially better results. This observation highlights the potential challenges these models may encounter when applied in contexts where the training dataset is relatively limited, as is the case for this problem.

Table 9: **Temperature Prediction: Pool Boiling Subcooled.**

|  | UNet$_{bench}$ | UNet$_{mod}$ | T-FNO | F-FNO | G-FNO | FNO | UNO |
|---|---|---|---|---|---|---|---|
| Rel. Err. | **0.036** | 0.051 | 0.052 | 0.050 | 0.064 | 0.070 | 0.062 |
| RMSE | **0.035** | 0.049 | 0.050 | 0.048 | 0.062 | 0.068 | 0.059 |
| BRMSE | **0.073** | 0.146 | 0.118 | 0.124 | 0.142 | 0.149 | 0.145 |
| IRMSE | **0.113** | 0.157 | 0.189 | 0.155 | 0.236 | 0.269 | 0.220 |
| Max Err. | 2.1 | **1.937** | 2.32 | 2.21 | 2.825 | 2.806 | 2.659 |
| F. Low | **0.281** | 0.348 | 0.373 | 0.472 | 0.507 | 0.625 | 0.440 |
| F. Mid | **0.266** | 0.386 | 0.390 | 0.375 | 0.519 | 0.559 | 0.453 |
| F. High | **0.040** | 0.059 | 0.055 | 0.057 | 0.061 | 0.063 | 0.068 |

Table 10: **Temperature Prediction: Pool Boiling Saturated.**

|  | UNet$_{bench}$ | UNet$_{mod}$ | T-FNO | F-FNO | G-FNO | FNO | UNO |
|---|---|---|---|---|---|---|---|
| Rel. Err. | **0.035** | 0.039 | 0.052 | 0.053 | 0.066 | 0.078 | 0.072 |
| RMSE | **0.035** | 0.039 | 0.052 | 0.052 | 0.065 | 0.076 | 0.071 |
| BRMSE | **0.082** | 0.147 | 0.165 | 0.163 | 0.218 | 0.214 | 0.214 |
| IRMSE | **0.052** | 0.083 | 0.125 | 0.095 | 0.152 | 0.140 | 0.116 |
| Max Err. | **1.701** | 1.785 | 2.055 | 1.788 | 1.793 | 2.106 | 2.23 |
| F. Low | 0.257 | **0.241** | 0.373 | 0.416 | 0.463 | 0.633 | 0.568 |
| F. Mid | **0.268** | 0.273 | 0.377 | 0.399 | 0.496 | 0.606 | 0.532 |
| F. High | **0.023** | 0.042 | 0.053 | 0.052 | 0.070 | 0.062 | 0.069 |

Table 11: **Temperature Prediction: Pool Boiling Gravity.**

|  | UNet$_{bench}$ | UNet$_{mod}$ | T-FNO | F-FNO | G-FNO | FNO | UNO |
|---|---|---|---|---|---|---|---|
| Rel. Err. | **0.042** | 0.051 | 0.062 | 0.061 | 0.103 | 0.104 | 0.081 |
| RMSE | **0.040** | 0.049 | 0.059 | 0.058 | 0.098 | 0.099 | 0.077 |
| BRMSE | **0.070** | 0.139 | 0.150 | 0.131 | 0.188 | 0.209 | 0.184 |
| IRMSE | **0.108** | 0.201 | 0.266 | 0.239 | 0.412 | 0.457 | 0.335 |
| Max Err. | 2.92 | **2.846** | 3.626 | 3.22 | 3.307 | 3.946 | 3.667 |
| F. Low | 0.491 | **0.466** | 0.500 | 0.578 | 1.035 | 1.103 | 0.746 |
| F. Mid | **0.275** | 0.326 | 0.440 | 0.423 | 0.704 | 0.724 | 0.548 |
| F. High | **0.033** | 0.049 | 0.054 | 0.049 | 0.061 | 0.067 | 0.077 |

## C.5 Learning Flow Dynamics Results

Table 14 shows the predicted temperature, $x$ velocity, and $y$ velocity during rollout of 200 timesteps on the pool boiling subcooled dataset. We observe an interesting phenomenon: UNet$_{bench}$ goes from

Table 12: **Temperature Prediction: Flow Boiling Gravity.**

|          | UNet$_{bench}$ | UNet$_{mod}$ | T-FNO | UNO |
|----------|----------|----------|-------|-----|
| Rel. Err. | **0.055** | 0.095 | 0.134 | 0.157 |
| RMSE | **0.051** | 0.088 | 0.123 | 0.144 |
| BRMSE | **0.080** | 0.240 | 0.267 | 0.301 |
| IRMSE | **0.091** | 0.214 | 0.351 | 0.379 |
| Max Err. | **2.560** | 2.800 | 3.951 | 3.990 |
| F. Low | **0.263** | 0.388 | 0.591 | 0.602 |
| F. Mid | **0.281** | 0.419 | 0.551 | 0.692 |
| F. High | **0.149** | 0.276 | 0.389 | 0.449 |

Table 13: **Temperature Prediction: Flow Boiling Inlet Velocity.** All models have substantially higher error compared to the other datasets. This is likely caused by the higher velocity. Since the temporal discretization is quite coarse, each bubble will take a large "jump" between frames that is difficult to capture accurately.

|          | UNet$_{bench}$ | UNet$_{mod}$ | T-FNO | UNO |
|----------|----------|----------|-------|-----|
| Rel. Err. | **0.106** | 0.122 | 0.202 | 0.225 |
| RMSE | **0.097** | 0.111 | 0.184 | 0.205 |
| BRMSE | **0.271** | 0.315 | 0.486 | 0.471 |
| IRMSE | **0.178** | 0.193 | 0.252 | 0.319 |
| Max Err. | 2.900 | **2.862** | 3.527 | 3.992 |
| F. Low | **0.571** | 0.584 | 1.061 | 0.981 |
| F. Mid | **0.511** | 0.566 | 1.084 | 1.334 |
| F. High | **0.264** | 0.315 | 0.494 | 0.523 |

being the best model at predicting temperature, to being among the worst. This is the simplest model, so it is understandable that its limited capacity prevents it from learning more complex functions. In comparison, UNet$_{mod}$ performs incredibly well. T-FNO and F-FNO also continue to very well. The best performing model is P-UNet$_{mod}$, which was trained using the pushforward trick. It achieves the best score in all metrics, except, surprisingly, for the maximum error. UNO experiences some divergence. Initially, it makes good predictions (perhaps better than the UNet architectures), but over long rollouts, its predictions tend toward infinity. Since UNO is the model that experiences the most striking divergence, we compare its 1D radially averaged power spectrum with UNet$_{mod}$ and P-UNet$_{mod}$ in Figure 10. We are able to reproduce findings that show that autoregressive Fourier models may experience aliasing errors, causing them to diverge [56].

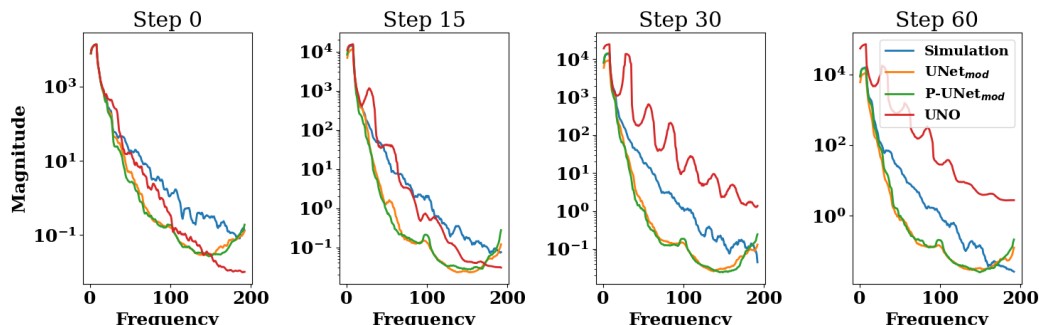

Figure 10: 1D Radially Averaged Power Spectrum of the simulation temperature and model rollout temperature. We see that initially, all of the models follow the simulations frequency fairly well, but UNO quickly diverges. This phenomena has been reported [56] as being aliasing errors. The plots are on log-scale, so curves below the simulation's are more accurate than curves above the simulation.

Table 14: **Coupled Velocity and Temperature Prediction: Pool Boiling Subcooled**$_{0.1}$. P-UNet$_{mod}$ was trained using the pushforward trick. It gets the best results during rollout. Notably, F-FNO performs better than UNet$_{bench}$ and UNet$_{mod}$ . This is a sharp contrast to the temperature predictions, where the UNet variants were far better. UNO diverged during rollout. This is likely because is mostly uses low frequencies. The error in the high frequencies accumulates during rollout, and it diverges.

|         | Metric   | UNet$_{bench}$ | UNet$_{mod}$ | P-UNet$_{mod}$ | T-FNO | UNO   | F-FNO |
|---------|----------|----------------|--------------|----------------|-------|-------|-------|
| Temp.   | Rel. Err. | 0.108         | 0.074        | **0.040**      | 0.066 | 0.322 | 0.067 |
|         | RMSE     | 0.105          | 0.072        | **0.039**      | 0.064 | 0.051 | 0.065 |
|         | BRMSE    | 0.248          | 0.211        | **0.113**      | 0.142 | 0.050 | 0.237 |
|         | IRMSE    | 0.178          | 0.191        | **0.129**      | 0.212 | 0.077 | 0.201 |
|         | Max Err. | 2.83           | **2.340**    | 2.741          | 2.566 | 7.492 | 2.488 |
|         | F. Low   | 1.407          | 0.495        | **0.244**      | 0.531 | 4.902 | 0.498 |
|         | F. Mid   | 0.740          | 0.583        | **0.280**      | 0.518 | 1.594 | 0.507 |
|         | F. High  | 0.090          | 0.076        | **0.051**      | 0.061 | 0.442 | 0.083 |
| $x$ Vel. | Rel. Err. | 0.681         | 0.553        | **0.429**      | 0.642 | 1.834 | 0.547 |
|         | RMSE     | 0.021          | 0.015        | **0.012**      | 0.018 | 0.051 | 0.015 |
|         | BRMSE    | 0.023          | 0.023        | **0.022**      | 0.023 | 0.050 | 0.024 |
|         | IRMSE    | 0.051          | 0.049        | **0.043**      | 0.049 | 0.077 | 0.050 |
|         | Max Err. | **0.294**      | 0.320        | 0.353          | 0.32  | 0.574 | 0.334 |
|         | F. Low   | 0.329          | 0.225        | **0.142**      | 0.258 | 0.677 | 0.189 |
|         | F. Mid   | 0.115          | 0.105        | **0.079**      | 0.110 | 0.328 | 0.107 |
|         | F. High  | 0.012          | 0.012        | **0.011**      | 0.012 | 0.092 | 0.012 |
| $y$ Vel. | Rel. Err. | 0.563         | 0.400        | **0.311**      | 0.415 | 1.170 | 0.397 |
|         | RMSE     | 0.021          | 0.015        | **0.012**      | 0.016 | 0.044 | 0.015 |
|         | BRMSE    | 0.015          | 0.012        | **0.010**      | 0.011 | 0.035 | 0.013 |
|         | IRMSE    | 0.046          | 0.046        | **0.038**      | 0.045 | 0.066 | 0.047 |
|         | Max Err. | 0.292          | **0.272**    | 0.280          | 0.290 | 0.288 | 0.289 |
|         | F. Low   | 0.387          | 0.190        | **0.132**      | 0.180 | 0.660 | 0.176 |
|         | F. Mid   | 0.128          | 0.112        | **0.085**      | 0.111 | 0.240 | 0.106 |
|         | F. High  | 0.011          | 0.010        | **0.009**      | 0.010 | 0.067 | 0.010 |

# D  Physical and non-dimensional values

## D.1  Fluid properties

Table 15 provides a comprehensive list of physical properties relevant to multiphase flow. These properties are presented with their respective symbols, units, and descriptions. It is important to consider the properties of the liquid and vapor phases separately in multiphase flows, as they exhibit distinct characteristics due to the phase change phenomenon. Additionally, the temperatures of the heater and liquid are also taken into account, as they significantly influence bubble dynamics and flow behaviors. The saturation temperature, which marks the initiation of bubble generation, is determined by the specific fluid being studied.

Table 15: Physical properties with symbol and unit

| Symbol | Unit | Description |
|---|---|---|
| $\rho_l$ | $kg/m^3$ | Liquid density |
| $\rho_v$ | $kg/m^3$ | Vapor density |
| $\mu_l$ | $N \cdot s/m^2$ | Dynamic viscosity of liquid |
| $\mu_v$ | $N \cdot s/m^2$ | Dynamic viscosity of vapor |
| $C_{p,l}$ | $J/kg \cdot K$ | Specific heat capacity of liquid |
| $C_{p,v}$ | $J/kg \cdot K$ | Specific heat capacity of vapor |
| $k_l$ | $W/m \cdot K$ | Thermal conductivity of liquid |
| $k_v$ | $W/m \cdot K$ | Thermal conductivity of vapor |
| $h_{vl}$ | $J/kg$ | Latent heat |
| $g$ | $m/s^2$ | Gravitational acceleration |
| $\sigma$ | $N/m$ | Surface tension |
| $T_{wall}$ | $K$ | Heater temperature |
| $T_{bulk}$ | $K$ | Bulk temperature |
| $T_{sat}$ | $K$ | Saturation temperature |

## D.2  Conversion to Non-dimensional parameters

Table 16 provides a comprehensive list of non-dimensional variables used in this study, including their symbols, definitions, and descriptions. For example, the non-dimensional time is obtained by dividing the characteristic length scale by the characteristic velocity scale. These non-dimensional variables are necessary to solve the non-dimensionalized governing equations such as the continuity, momentum, and energy equations. Additionally, non-dimensional properties such as density, dynamic viscosity, specific heat capacity, and thermal conductivity are also considered in the context of multiphase flow. Notably, representative parameters in fluid mechanics, thermodynamics, and heat transfer, such as the Reynolds number ($Re$), Froude number ($Fr$), Prandtl number ($Pr$), Stefan number ($St$), Weber number ($We$), and Peclet number ($Pe$), are used to obtain simulation results. By employing non-dimensionalization, the effects of different physical quantities can be studied independently of their specific units, facilitating a deeper understanding of the underlying phenomena.

# E  Simulation Details

## E.1  Multiphase Simulations

Numerical simulations of multiphase flows with phase changes have been studied using various techniques to model the behavior at the liquid-vapor interface and track its evolution over time. Two commonly used methods for handling boundary conditions related to surface tension and evaporation are the ghost fluid method (GFM) and the continuum surface force method (CSF). The GFM enforces a sharp jump in pressure, velocity, and temperature across the interface, while the CSF diffuses the forcing within the vicinity of the interface for a smoother transition. The choice between these approaches involves a trade-off between accuracy and stability, with GFM offering higher accuracy but lower stability compared to CSF. Interface tracking is typically achieved implicitly using level-set or volume of fluid (VOF) techniques.

Table 16: Formulae used for conversion of real world values to non-dimensional values

| Symbol | Definition | Description |
|--------|------------|-------------|
| $\rho^*$ | $\rho_v/\rho_l$ | Non-dimensional density |
| $\mu^*$ | $\mu_v/\mu_l$ | Non-dimensional viscosity |
| $C_p^*$ | $C_{p,v}/C_{p,l}$ | Non-dimensional specific heat capacity |
| $k^*$ | $k_v/k_l$ | Non-dimensional thermal conductivity |
| $l_0$ | $\sqrt{\sigma/g\Delta\rho}$ | Characteristic length scale |
| $u_0$ | $\sqrt{gl_0}$ | Characteristic velocity scale |
| $t_0$ | $l_0/u_0$ | Characteristic time scale |
| $T_{wall}^*$ | $(T_{wall} - T_{bulk})/(T_{wall} - T_{bulk}) = 1$ | Non-dimensional heater temperature |
| $T_{bulk}^*$ | $(T_{bulk} - T_{bulk})/(T_{wall} - T_{bulk}) = 0$ | Non-dimensional bulk temperature |
| $T_{sat}^*$ | $(T_{sat} - T_{bulk})/(T_{wall} - T_{bulk})$ | Non-dimensional saturated temperature |
| $Re$ | $\rho_l u_0 l_0/\mu_l$ | Reynolds number |
| $Fr$ | $u_0/\sqrt{gl_0}$ | Froude number |
| $Pr$ | $\mu_l C_{p,l}/k_l$ | Prandtl number |
| $St$ | $C_{p,l}(T_{wall} - Tbulk)/h_{vl}$ | Stefan number |
| $We$ | $\rho_l u_0^2 l_0/\sigma$ | Weber number |
| $Pe$ | $Re \cdot Pr$ | Peclet number |

Researchers have employed these methods to study and model various aspects of multiphase flows with phase changes. For example, Gibou et al. [75] used a level-set method with sharp interfacial jump conditions within the framework of the GFM to model homogeneous two-dimensional evaporation and film boiling. Son and Dhir [76, 77] extended this approach to perform heterogeneous pool boiling calculations involving single and multiple bubbles. Majority of their initial work focused on model development and verification using two-dimensional (2D) simulations, since real world three-dimensional (3D) calculations were expensive due to limitations of the software framework.

Efforts have also been made to perform high-fidelity 3D simulations of pool boiling by combining different techniques. Yazadani et al. [78] conducted critical heat flux (CHF) calculations on earth gravity using a combination of VOF and CSF methods. Sato et al. [79, 11], on the other hand used the level-set method in combination with CSF for their simulations. These studies highlighted the computational cost associated with boiling simulations which had to be mitigated by performing low resolution calculations. More recently, Dhruv et al. [31, 32] used a combination of level-set and GFM methods for gravity scaling analysis of pool boiling at finer resolution than previous studies using adaptive mesh refinement (AMR) within the framework of FLASH. These simulations were applied to study effects of gravity on boiling heat transfer which lead to verification of experiment based heat flux models and enabled the quantification of turbulent heat flux associated with bubble dynamics during bubbly and slug flow [32]. The implementation of multiphase models within FLASH has transitioned to Flash-X, which leverages state-of-art AMR techniques and heterogeneous supercomputing architectures to significantly improve performance of boiling calculations. The BubbleML dataset is curated from simulations carried out with the Flash-X framework. Figures 11 and 12 illustrate the boundary conditions and describe the parameters of the simulations comprising the dataset.

An important note is that simulations still heavily rely on experimental observations to determine input conditions such as nucleation site distribution, bubble nucleation frequency, and solid-liquid-vapor contact angle dynamics [31]. As a result, simulations serve as an effective tool to understand and quantify trends in boiling regimes, rather than attempting to replicate experiments precisely. This opens up an opportunity for the integration of data-driven ML techniques, which can leverage diverse datasets to make informed predictions for boiling phenomena.

### E.2 Fluid parameters

The input configuration file of a simulation requires the inclusion of certain parameters, which remain constant for a specific fluid. For our simulations using FC-72 (Perfluorohexane, $C_6F_{14}$), the non-dimensional values for this fluid are provided in Table 17. It is important to note that parameters corresponding to any specific real-world fluid must be converted to these non-dimensional values

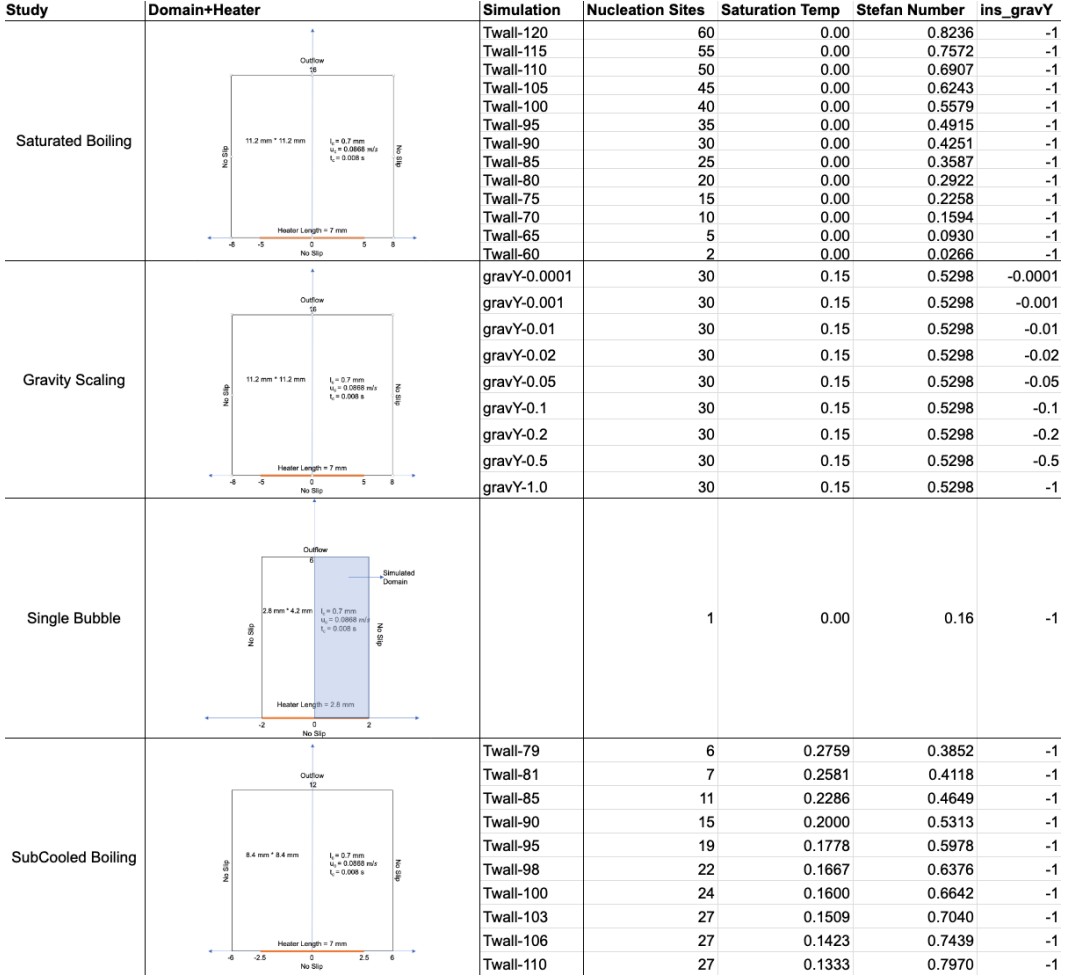

| Study | Domain+Heater | Simulation | Nucleation Sites | Saturation Temp | Stefan Number | ins_gravY |
|---|---|---|---|---|---|---|
| Saturated Boiling | | Twall-120 | 60 | 0.00 | 0.8236 | -1 |
| | | Twall-115 | 55 | 0.00 | 0.7572 | -1 |
| | | Twall-110 | 50 | 0.00 | 0.6907 | -1 |
| | | Twall-105 | 45 | 0.00 | 0.6243 | -1 |
| | | Twall-100 | 40 | 0.00 | 0.5579 | -1 |
| | | Twall-95 | 35 | 0.00 | 0.4915 | -1 |
| | | Twall-90 | 30 | 0.00 | 0.4251 | -1 |
| | | Twall-85 | 25 | 0.00 | 0.3587 | -1 |
| | | Twall-80 | 20 | 0.00 | 0.2922 | -1 |
| | | Twall-75 | 15 | 0.00 | 0.2258 | -1 |
| | | Twall-70 | 10 | 0.00 | 0.1594 | -1 |
| | | Twall-65 | 5 | 0.00 | 0.0930 | -1 |
| | | Twall-60 | 2 | 0.00 | 0.0266 | -1 |
| Gravity Scaling | | gravY-0.0001 | 30 | 0.15 | 0.5298 | -0.0001 |
| | | gravY-0.001 | 30 | 0.15 | 0.5298 | -0.001 |
| | | gravY-0.01 | 30 | 0.15 | 0.5298 | -0.01 |
| | | gravY-0.02 | 30 | 0.15 | 0.5298 | -0.02 |
| | | gravY-0.05 | 30 | 0.15 | 0.5298 | -0.05 |
| | | gravY-0.1 | 30 | 0.15 | 0.5298 | -0.1 |
| | | gravY-0.2 | 30 | 0.15 | 0.5298 | -0.2 |
| | | gravY-0.5 | 30 | 0.15 | 0.5298 | -0.5 |
| | | gravY-1.0 | 30 | 0.15 | 0.5298 | -1 |
| Single Bubble | | | 1 | 0.00 | 0.16 | -1 |
| SubCooled Boiling | | Twall-79 | 6 | 0.2759 | 0.3852 | -1 |
| | | Twall-81 | 7 | 0.2581 | 0.4118 | -1 |
| | | Twall-85 | 11 | 0.2286 | 0.4649 | -1 |
| | | Twall-90 | 15 | 0.2000 | 0.5313 | -1 |
| | | Twall-95 | 19 | 0.1778 | 0.5978 | -1 |
| | | Twall-98 | 22 | 0.1667 | 0.6376 | -1 |
| | | Twall-100 | 24 | 0.1600 | 0.6642 | -1 |
| | | Twall-103 | 27 | 0.1509 | 0.7040 | -1 |
| | | Twall-106 | 27 | 0.1423 | 0.7439 | -1 |
| | | Twall-110 | 27 | 0.1333 | 0.7970 | -1 |

Figure 11: Parameters for 2D pool boiling simulations

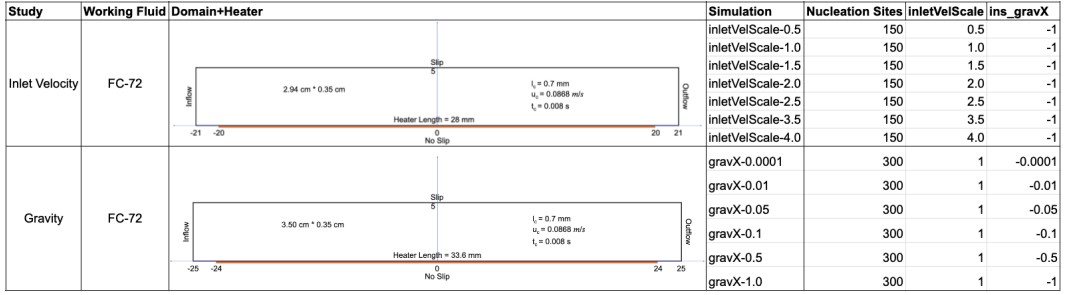

| Study | Working Fluid | Domain+Heater | Simulation | Nucleation Sites | inletVelScale | ins_gravX |
|---|---|---|---|---|---|---|
| Inlet Velocity | FC-72 | | inletVelScale-0.5 | 150 | 0.5 | -1 |
| | | | inletVelScale-1.0 | 150 | 1.0 | -1 |
| | | | inletVelScale-1.5 | 150 | 1.5 | -1 |
| | | | inletVelScale-2.0 | 150 | 2.0 | -1 |
| | | | inletVelScale-2.5 | 150 | 2.5 | -1 |
| | | | inletVelScale-3.5 | 150 | 3.5 | -1 |
| | | | inletVelScale-4.0 | 150 | 4.0 | -1 |
| Gravity | FC-72 | | gravX-0.0001 | 300 | 1 | -0.0001 |
| | | | gravX-0.01 | 300 | 1 | -0.01 |
| | | | gravX-0.05 | 300 | 1 | -0.05 |
| | | | gravX-0.1 | 300 | 1 | -0.1 |
| | | | gravX-0.5 | 300 | 1 | -0.5 |
| | | | gravX-1.0 | 300 | 1 | -1 |

Figure 12: Parameters for 2D flow boiling simulations

before being input into the simulation configuration files. This conversion allows for consistent and standardized representation of fluid properties in the simulation, enabling accurate and meaningful results to be obtained.

Table 17: Non-dimensional constants for FC-72, the fluid used in BubbleML

| Parameter | Variable Name | Non-dimensional Value |
|---|---|---|
| Inverse Reynolds Number, $\frac{1}{Re}$ | ins_invReynolds | 0.0042 |
| Non-dimensional density, $\rho^*$ | mph_rhoGas | 0.0083 |
| Non-dimensional viscosity, $\mu^*$ | mph_muGas | 1.0 |
| Non-dimensional thermal conductivity, $k^*$ | mph_thcoGas | 0.25 |
| Non-dimensional specific heat capacity, $C_p^*$ | mph_CpGas | 0.83 |
| Inverse Weber Number, $\frac{1}{We}$ | mph_invWeber | 1.0 |
| Prandtl Number, $Pr$ | ht_Prandtl | 8.4 |

