# F BubbleML Datasheet

In this section, we provide details about the BubbleML dataset published with a permanent doi 10.5281/zenodo.8039786.

## F.1 Motivation

1. **For what purpose was the dataset created?** Was there a specific task in mind? Was there a specific gap that needed to be filled? Please provide a description.

   The dataset was created to enable machine learning driven research on phase change phenomena by providing a comprehensive, open source collection of high fidelity Boiling Simulations.

2. **Who created this dataset (e.g. which team, research group) and on behalf of which entity (e.g., company, institution, organization)?**

   The dataset was primarily created by Sheikh Md Shakeel Hassan, Arthur Feeney (PhD students at HPCForge, University of California Irvine) in collaboration with Akash Dhruv(Argonne National Laboratory) and Jihoon Kim(Korea University).

3. **Who funded the creation of the dataset?** If there is an associated grant, please provide the name of the grant or and the grant name and number.)

   The work was funded by the National Science Foundation (NSF) under the award number 1750549.

4. **Any other comments?**

   No.

## F.2 Composition

1. **What do the instances that comprise the dataset represent (e.g. documents, photos, people, countries)?** Are there multiple types of instances (e.g. movies, users, and ratings; people and interactions between them; nodes and edges)? Please provide a description.)

   Each instance of the dataset represents a study of pool boiling or flow boiling where a particular parameter such as heater temperature or gravity is varied over a range of values.

2. **How many instances are there in total (of each type, if appropriate)?**

   There are a total of 6 boiling studies done in 2D, comprising of 4 pool boiling and 2 flow boiling studies. The dataset also contains 2 3D pool boiling simulations. For details of each study please refer Table 1.

3. **Does the dataset contain all possible instances or is it a sample(not necessarily random) of instances from a larger set?** If the dataset is a sample, then what is the larger set? Is the sample representative of the larger set (e.g. geographic coverage)? If so, please describe how this representativeness was validated/verified. If it is not representative of the larger set, please describe why not (e.g., to cover a more diverse range of instances, because instances were withheld or unavailable).)

   It is practically impossible to cover all possible instances of Multiphase Simulations. Our dataset encompasses a wide range of physical phenomena that can be seen in boiling fluids. Nevertheless, we provide a framework for new data generation 5 using the same tools that we have used in this paper.

4. **What data does each instance consist of?** ("Raw" data (e.g. unprocessed text or images) or features? In either case, please provide a description.)

   Each instance consists of a number of hdf5 simulation files. Each such file contains velocity, temperature, pressure and liquid-vapor phase information in numpy arrays of dimensions $(t, x, y)$.

5. **Is there a label or target associated with each instance?** If so, please provide a description.

   Each simulation is associated with a set of physical runtime parameters. They are included in each hdf5 simulation file and can be used for training purposes.

6. **Is any information missing from individual instances?** (If so, please provide a description, explaining why this information is missing (e.g., because it was unavailable). This does not include intentionally removed information, but might include, e.g., redacted text.)

No.

7. **Are relationships between individual instances made explicit (e.g., users' movie ratings, social network links)?** (If so, please describe how these relationships are made explicit.)

   The studies are independent from each other.

8. **Are there recommended data splits (e.g., training, development/validation, testing)?** (If so, please provide a description of these splits, explaining the rationale behind them.)

   The training, validation and testing splits can be at the end-users discretion depending on the downstream task the dataset is used for.

9. **Are there any errors, sources of noise, or redundancies in the dataset?** (If so, please provide a description.)

   Small discretization errors are inevitable in numerical simulations. However, the simulations capture the trends seen in experimental data and the errors do not affect the correctness of the dataset.

10. **Is the dataset self-contained, or does it link to or otherwise rely on external resources (e.g., websites, tweets, other datasets)?** (If it links to or relies on external resources, a) are there guarantees that they will exist, and remain constant, over time; b) are there official archival versions of the complete dataset (i.e., including the external resources as they existed at the time the dataset was created); c) are there any restrictions (e.g., licenses, fees) associated with any of the external resources that might apply to a future user? Please provide descriptions of all external resources and any restrictions associated with them, as well as links or other access points, as appropriate.)

    Yes.

11. **Does the dataset contain data that might be considered confidential (e.g., data that is protected by legal privilege or by doctor-patient confidentiality, data that includes the content of individuals' non-public communications)?** (If so, please provide a description.)

    No.

12. **Does the dataset contain data that, if viewed directly, might be offensive, insulting, threatening, or might otherwise cause anxiety?** (If so, please describe why.)

    No.

13. **Does the dataset relate to people?** (If not, you may skip the remaining questions in this section.)

    No.

14. **Does the dataset identify any subpopulations (e.g., by age, gender)?** (If so, please describe how these sub-populations are identified and provide a description of their respective distributions within the dataset.)

    N/A.

15. **Is it possible to identify individuals (i.e., one or more natural persons), either directly or indirectly (i.e., in combination with other data) from the dataset?** (If so, please describe how.)

    N/A.

16. **Does the dataset contain data that might be considered sensitive in any way (e.g., data that reveals racial or ethnic origins, sexual orientations, religious beliefs, political opinions or union memberships, or locations; financial or health data; biometric or genetic data; forms of government identification, such as social security numbers; criminal history)?** (If so, please provide a description.)

    N/A.

17. **Any other comments?**

    No.

### F.3   Collection Process

1. **How was the data associated with each instance acquired?** (Was the data directly observable (e.g., raw text, movie ratings), reported by subjects (e.g., survey responses), or indirectly inferred/derived from other data (e.g., part-of-speech tags, model-based guesses

for age or language)? If data was reported by subjects or indirectly inferred/derived from other data, was the data validated/verified? If so, please describe how.)

The multiphase simulations were performed using Flash-X[21]. Some example configuration files for the simulations provided in the dataset can be found here[5]. The simulations have been validated against real world experimental trends previously established in literature. Please refer to section A.4 for details.

2. **What mechanisms or procedures were used to collect the data(e.g.,hardware apparatus or sensor, manual human curation, software program, software API)?** (How were these mechanisms or procedures validated?)

The simulations were run on the HPC3 cluster in University of California Irvine.

3. **If the dataset is a sample from a larger set, what was the sampling strategy (e.g., deterministic, probabilistic with specific sampling probabilities)?**

Since it is impossible to capture all possible physical conditions of a Multiphysics system, studies were selected to capture a wide range of physical phenomena such as varying heater temperature, gravity and flow velocity.

4. **Who was involved in the data collection process (e.g., students, crowd workers, contractors) and how were they compensated (e.g., how much were crowd workers paid)?**

Data generation was done by the authors.

5. **Over what timeframe was the data collected?** (Does this timeframe match the creation timeframe of the data associated with the instances (e.g., recent crawl of old news articles)? If not, please describe the timeframe in which the data associated with the instances was created.)

Simulations were created over the time period April-June 2023.

6. **Were any ethical review processes conducted (e.g., by an institutional review board)?** (If so, please provide a description of these review processes, including the outcomes, as well as a link or other access point to any supporting documentation.)

No.

7. **Does the dataset relate to people?** (If not, you may skip the remaining questions in this section.)

No.

8. **Did you collect the data from the individuals in question directly, or obtain it via third parties or other sources (e.g., websites)?**

N/A.

9. **Were the individuals in question notified about the data collection?** (If so, please describe (or show with screenshots or other information) how notice was provided, and provide a link or other access point to, or otherwise reproduce, the exact language of the notification itself.)

N/A.

10. **Did the individuals in question consent to the collection and use of their data?** (If so, please describe (or show with screenshots or other information) how consent was requested and provided, and provide a link or other access point to, or otherwise reproduce, the exact language to which the individuals consented.)

N/A.

11. **If consent was obtained, were the consenting individuals provided with a mechanism to revoke their consent in the future or for certain uses?** (If so, please provide a description, as well as a link or other access point to the mechanism (if appropriate).)

N/A.

12. **Has an analysis of the potential impact of the dataset and its use on data subjects (e.g., a data protection impact analysis) been conducted?** (If so, please provide a description of this analysis, including the outcomes, as well as a link or other access point to any supporting documentation.)

N/A.

13. **Any other comments?**

None.

### F.4 Preprocessing/cleaning/labeling

1. **Was any preprocessing/cleaning/labeling of the data done(e.g.,discretization or bucketing, tokenization, part-of-speech tagging, SIFT feature extraction, removal of instances, processing of missing values)?** (If so, please provide a description. If not, you may skip the remainder of the questions in this section.)

   The raw data obtained from the simulations are in a block structured format for efficient memory management in 3D simulations. For 2D simulations, the block structure can be abandoned as a single unblocked simulation is only 2-3 GB in size. This makes data access easier. We provide the 2D simulations in an unblocked format as they are easier to read and load into existing PyTorch and Tensorflow workflows.

2. **Was the "raw" data saved in addition to the preprocessed/cleaned/labeled data (e.g., to support unanticipated future uses)?** (If so, please provide a link or other access point to the "raw" data.)

   The raw simulation data is saved on the cluster used to run the simulations. However we do not anticipate any future uses for the raw data as the data contained in the unblocked simulations is essentially the same, but the raw unblocked data can be made available upon request

3. **Is the software used to preprocess/clean/label the instances available?** (If so, please provide a link or other access point.)

   The preprocessing code is available in the github repository[3].

4. **Any other comments?**

   None.

### F.5 Uses

1. **Has the dataset been used for any tasks already?** (If so, please provide a description.)

   The 3D simulations have been used in previous papers which have been appropriately cited in this paper and a few papers are a work in progress.

2. **Is there a repository that links to any or all papers or systems that use the dataset?** (If so, please provide a link or other access point.)

   Not currently.

3. **What (other) tasks could the dataset be used for?**

   The dataset is primarily intended for the scientific machine learning community. The dataset could probably be used for other studies such as bubble segmentation. We invite the community to use the dataset in innovative ways.

4. **Is there anything about the composition of the dataset or the way it was collected and preprocessed/cleaned/labeled that might impact future uses?** (For example, is there anything that a future user might need to know to avoid uses that could result in unfair treatment of individuals or groups (e.g., stereotyping, quality of service issues) or other undesirable harms (e.g., financial harms, legal risks) If so, please provide a description. Is there anything a future user could do to mitigate these undesirable harms?)

   No.

5. **Are there tasks for which the dataset should not be used?** (If so, please provide a description.)

   No.

6. **Any other comments?**

   None.

### F.6 Distribution

1. **Will the dataset be distributed to third parties outside of the entity (e.g., company, institution, organization) on behalf of which the dataset was created?** (If so, please provide a description.)

   Yes, the dataset is freely and publicly available and accessible.

2. **How will the dataset will be distributed (e.g., tarball on website, API, GitHub)?** (Does the dataset have a digital object identifier (DOI)?)

   The dataset is free for download by everyone. Links are available in the github repository[3]. The doi of the dataset is https://doi.org/10.5281/zenodo.8039786.

3. **When will the dataset be distributed?**

   The dataset is distributed as of June 2023 in its first version.

4. **Will the dataset be distributed under a copyright or other intellectual property (IP) license, and/or under applicable terms of use (ToU)?** (If so, please describe this license and/or ToU, and provide a link or other access point to, or otherwise reproduce, any relevant licensing terms or ToU, as well as any fees associated with these restrictions.)

   The dataset is licensed under CC BY license.

5. **Have any third parties imposed IP-based or other restrictions on the data associated with the instances?** (If so, please describe these restrictions, and provide a link or other access point to, or otherwise reproduce, any relevant licensing terms, as well as any fees associated with these restrictions.)

   No

6. **Do any export controls or other regulatory restrictions apply to the dataset or to individual instances?** (If so, please describe these restrictions, and provide a link or other access point to, or otherwise reproduce, any supporting documentation.)

   No

7. **Any other comments?**

   None.

## F.7 Maintenance

1. **Who is supporting/hosting/maintaining the dataset?**

   The dataset is being maintained by the High Performance Computing laboratory(HPCForge) of the EECS department in University of California Irvine.

2. **How can the owner/curator/manager of the dataset be contacted (e.g., email address)?**

   The manager of the dataset can be reached at `sheikhh1@uci.edu`.

3. **Is there an erratum?** (If so, please provide a link or other access point.)

   Currently, there is no erratum. If errors are encountered, the dataset will be updated with a fresh version. They will all be provided in the same github repository[3].

4. **Will the dataset be updated (e.g., to correct labeling errors, add new instances, delete instances')?** (If so, please describe how often, by whom, and how updates will be communicated to users (e.g., mailing list, GitHub)?)

   Same as above.

5. **If the dataset relates to people, are there applicable limits on the retention of the data associated with the instances (e.g., were individuals in question told that their data would be retained for a fixed period of time and then deleted)?** (If so, please describe these limits and explain how they will be enforced.)

   N/A.

6. **Will older versions of the dataset continue to be supported/hosted/maintained?** (If so, please describe how. If not, please describe how its obsolescence will be communicated to users.)

   Versioning of the dataset will be maintained in the github repository and the homepage.

7. **If others want to extend/augment/build on/contribute to the dataset, is there a mechanism for them to do so?** (If so, please provide a description. Will these contributions be validated/verified? If so, please describe how. If not, why not? Is there a process for communicating/distributing these contributions to other users? If so, please provide a description.)

   Flash-X is a publicly available software and simulations can be generated by anyone with the available compute resources. Examples are provided [5].

8. **Any other comments?**
   None.

## F.8  Reproducibility of the baseline score

The training details and models are provided in the BubbleML github repository[3] for reproducibility of reported results.

## F.9  Reading and using the dataset

The datasets are presented as compressed tarballs, one for each study. Upon decompressing, a number of HDF5 files are obtained which can be read by any standard library(e.g., h5py for Python). For code examples in python, please refer to the github repository[3]. Scripts are provided for loading the data into scientific machine learning models and creating Optical Flow datasets.

## F.10  Data Format

Each HDF5 file has the following keys: "dfun", "int-runtime-params", "pressure", "real-runtime-params", "temperature", "velx", "vely", "x" and "y". "dfun", "pressure", "temperature", "velx", "vely", "x" and "y" are numpy arrays of size (Temporal Dimension, X-Spatial Dimension, Y-Spatial Dimension). These keys contain the liquid-vapor phase information, pressure, temperature, velocities in x and y direction and the x and y coordinates in non-dimensional units respectively. The keys "int-runtime-params" and "real-runtime-params" contain various physical constants(Reynolds Number, Prandtl Number etc.) that were used in the respective simulations.