# OpenReview forum: "BubbleML: A Multiphase Multiphysics Dataset and Benchmarks for Machine Learning"
_NeurIPS.cc/2023/Track/Datasets_and_Benchmarks — NeurIPS 2023 Datasets and Benchmarks Spotlight_

### Official Review · Reviewer_5sS9 · 2023-07-20
**BubbleML: A Multi-Physics Dataset and Benchmarks for Machine Learning**

**Rating:** 7
**Confidence:** 4
**Clarity:** Yes, the paper is well written

**Strengths:**

- This paper provides a comprehensive dataset involving a wide range of two-phase (liquid-vapor) change phenomena in boiling, with a focus on bubble and flow dynamics.

- This paper performs validation against experimental data to ensure the dataset's accuracy and reliability.

- The proposed dataset BubbleML can facilitate diverse downstream tasks. In order to show this, the paper provides two benchmark scenarios: (1) optical flow for learning bubble dynamics and scientific machine learning using operator networks to learn temperature dynamics and estimate heatflux.

**Additional Feedback:**

See "Opportunities For Improvement"

**Correctness:**

The dataset construction seems to be solid, but more methods need to be included to make it a thorough benchmark. See "Opportunities For Improvement" for more detail.

**Documentation:**

Documentation is not enough. Please see "Opportunities For Improvement" for more details.

**Limitations:**

Limitation of this paper is not well discussed.

**Opportunities For Improvement:**

- On the github, I recommend authors provide a jupyter notebook to demonstrate how to load data, process data, train the model, and evaluate the performance, as this will be very useful for users to quickly use the dataset and benchmark. Also, I recommend authors put the benchmark result on the github readme as this is also easy for people to check the result.
- This paper only benchmarks three models, which is not enough for a benchmark. I recommend authors include the following models [1][2][3] in the benchmark to make it more aligned with the recent advances in PDE modeling.
- Does multi-phase and multi-physics refer to the same thing?
- In the paper, authors mentioned these three baselines are belongs to “operator network”, which makes me confused after I read some other related papers, such as [1]. According to the description of how “operator networks” works in the paper, I feel like they belong to the autoregressive way of solving PDE instead of the neural operator (of course FNO is a neural operator, but I guess authors use autoregressive prediction version of FNO).


[1] Brandstetter, Johannes, Daniel Worrall, and Max Welling. "Message passing neural PDE solvers." arXiv preprint arXiv:2202.03376 (2022).\
[2] Tran, Alasdair, et al. "Factorized fourier neural operators." arXiv preprint arXiv:2111.13802 (2021).\
[3] Helwig, Jacob, et al. "Group Equivariant Fourier Neural Operators for Partial Differential Equations." arXiv preprint arXiv:2306.05697 (2023).

**Relation To Prior Work:**

Yes

**Summary And Contributions:**

This paper proposes a dataset and benchmark targeting phase change phenomena. Specifically, they design a BubbleML dataset that leverages physics-driven simulations to provide accurate ground truth information for various boiling scenarios, including nucleate pool boiling, flow boiling, and sub-cooled boiling. This dataset covers a wide range of parameters, including varying gravity conditions, flow rates, sub-cooling levels, and wall superheat, comprising 51 simulations. They showcase the dataset’s potential to facilitate the exploration of diverse downstream tasks by introducing two benchmarks: (a) optical flow analysis to capture bubble dynamics, and (b) operator networks for learning temperature dynamics.

---

> ### Author Response · Authors · 2023-08-22
>
> We are grateful for the suggestions and have incorporated many of your suggestions into the paper and github repository.
>
> 1. We have added two self-contained jupyter notebooks to the github repository (https://github.com/HPCForge/BubbleML/tree/main/examples). One shows how to load the data using h5py, list out the keys, access the metadata, and visualize the different fields. The second notebook is a simplified version of our SciML training experiments. This shows the pipeline for autoregressive training and visualizations of the one-step errors. Both examples use a downsampled version of the Subcooled Pool Boiling dataset, which we keep in the examples/ directory for easy access.
>
> 2. We have included experiments for the push-forward trick, temporal bundling, Factorized FNO, and Group-Equivariant FNO from the papers you recommended. Each model has been evaluated on several studies. We have included a more thorough discussion of each of the models and training strategies in Appendix C1 and C2.
>
> 3. Multiphase and multiphysics refer to different concepts. In the context of BubbleML, “multiphase” refers to the modeling of liquid-gas interactions and “multiphysics” refers to the system of equations that are solved to model multiphase heat transfer physics. Simulating heat transfer in a fluid flow requires solving both the Navier-Stokes equations for fluid dynamics and the heat transfer equation (i.e., energy equation) simultaneously.
>
> 4. We have improved our description of neural operators in Section 4.2. Normally, a neural operator would map some initial timesteps to all of the remaining ones. However, this is not practical in cases when the simulation requires reaching a quasi-steady state. It’s more effective to generate a few simulations with many timesteps, rather than many simulations with a few timesteps. This essentially requires using models in an auto-regressive form. We have pointed to several papers that use neural operators this way.
>
> 5. We have also added a discussion of limitations to the conclusions, and added discussions of “open problems” to the Optical Flow and SciML Sections 4.1 and 4.2.
>
> 6. Finally, in addition to adding the examples, we have added documentation of the dataset that describes each of the fields, the memory layout of the data, and how to access important metadata, such as the Reynolds number or variables used to compute thermal diffusivity. It also discusses potential pitfalls with non-dimensionalization and steady-state. (https://github.com/HPCForge/BubbleML/blob/main/bubbleml_data/DOCS.md)

---

> > ### Comment · Reviewer_5sS9 · 2023-08-24
> >
> > My concerns are well addressed, and I increase my score to 7.

---

### Official Review · Reviewer_obr7 · 2023-07-20

**Rating:** 8
**Confidence:** 4
**Correctness:** Yes
**Clarity:** Yes

**Strengths:**

1. The datasets are clearly motivated in actual cooling cases.
2. The correctness of the simulation is tested.
3. The evaluation tasks are meaningful in the real world and the experiments are extensive.

**Additional Feedback:**

N/A

**Documentation:**

The documentation is clear, and the running samples and checkpoints are available.

**Opportunities For Improvement:**

1. This URL is not available: https://hpcforge.eng.uci.edu/project/bubbleML
2. Performance of more models could be included. It will be better to show what will be the negative impact if the error prediction is not accurate enough. Is the accuracy of current models high enough?

**Relation To Prior Work:**

Yes

**Summary And Contributions:**

The authors provided datasets for boiling processes with diverse datasets and evaluation tasks. The motivation is clear, and the simulation is meaningful in the real world. The generation process is well-illustrated, and the datasets and codes are well-documented.

---

> ### Author Response · Authors · 2023-08-22
>
> We thank the reviewer for their valuable comments.
>
> The website link which was previously not working has been updated to https://hpcforge.github.io/BubbleML/ This has also been added to Appendix A.1.
>
> We have added several additional models to our SciML benchmarks. We refer the reviewers to updates in sections 4.2 and Appendix C where we have added results for F-FNO, G-FNO and models with push-forward trick. The need to improve current models is also discussed in the “Open Problems” subheading in Section 4.2. For optical flow, we have updated Sections 4.1 and Appendix B.3, especially improving on the error analysis that outlines the need for new models to improve the current levels of accuracy.

---

> > ### Comment · Reviewer_obr7 · 2023-08-27
> >
> > Thank you very much for the clarification! Thus, I increased my score.

---

### Official Review · Reviewer_EMa8 · 2023-07-20
**A dataset on boiling dynamics and benchmarks on downstream tasks**

**Rating:** 7
**Confidence:** 3

**Strengths:**

While there are plenty of PDE-related datasets, the one introduced in this work focuses on a more complex multiphase system, and it is verified against physics models from two aspects. A distinguishing aspect of the dataset is its consideration of various boundary conditions. The dataset has significant potential for impact, given the wide presence of boiling usages in real-life settings. Through experiments, the authors show that sub-datasets of BubbleML helps boost the performance of optical flow models. This illustrates the versatility of the dataset. A quick review of the accompanying GitHub repository confirms the presence of the dataset, experiment scripts, and guidance for reproducing results. The data sheet is also attached in the appendix.

**Additional Feedback:**

Overall, this submission could be enhanced further if the concerns raised are adequately addressed.

**Clarity:**

Figure 1 - as the most important figure, it has insufficient caption to describe what is happening in each part of (a), (b), and (c). It currently seems crammed with content.

Optical flow - it is understandable that most dynamical systems datasets with a temporal aspect can be used for this purpose. What makes BubbleML specially suited for this study?

The majority of the paper is well-written.

**Correctness:**

The authors claim that the dataset is corroborated by physics models, which is validated. The authors make only a few claims in the experiment section, mainly addressing performance of the models.

**Documentation:**

Yes, see appendix and code repository.

**Ethics:**

No significant concerns.

**Limitations:**

There is currently no content on the limitation of the dataset/benchmark in the main text. While not strictly necessary, it is encouraged so that users are aware of the limitations of the generated dataset. There are no significant potential negative social impact.

**Opportunities For Improvement:**

Since the submission has both a dataset and a benchmark section, it should be evaluated on both of these aspects:

- For the dataset, the authors provide qualitative justification for the choice of variables for simulation. However, the quantitative justification is missing, i.e. within Table 1, it is unclear what determines the number of simulations and their resolutions/time-steps, which makes the dataset settings seem arbitrary. This is not explained in the data sheet as well. In addition, the dataset does not come with the means of validation described in Sec. 3.4, considerably hindering the user’s ability to generate new datasets.

- For the benchmark(s), the submission is missing experiments that tests the current state-of-the-art models’ ability to model other aspects the boiling process, such as mass transfer and fluid dynamics. The heat transfer experiments should be complemented with other modeling studies. For heatflux studies, more discussion of the results is needed, i.e. what are the factors that affect performance of different methods. I would also suggest multiple random seeds to assess the noise within results.

**Relation To Prior Work:**

Yes, a clear distinction is drawn between BubbleML and other PDE datasets.

**Summary And Contributions:**

BubbleML consists of a multi-physics dataset and two separate set of experiments on optical flow and learning temperature dynamics. It is motivated by the lack of complex, high-fidelity physics dataset for studying phase changes. Generation of the dataset is done through simulation with Flash-X, with variations in physical parameters that control the system. For benchmarking, the paper presents some fine-tuning experiments on optical flow datasets and the study of heatflux evaluating three SciML baseline models.

---

> ### Author Response · Authors · 2023-08-22
>
> We thank the reviewer for their valuable comments to improve our paper. We have summarized our responses below and pointed to the updated sections in the paper below that address the specific concerns.
>
> 1. Opportunities for improvement, Q.1: For a quantitative justification of the choice of variables we have used in the simulation, we have added Appendix A.3.
>
> 2. Opportunities for improvement, Q.2: We have added benchmarks for learning fluid dynamics in Section 4.2 and Appendix C.5. In addition, we’ve included more discussion of the results including a subheading on “Open Problems”.
>
> 3. Clarity, 1: We have improved the caption for Figure 1 explaining the diverse physical phenomena seen in BubbleML.
>
> 4. Clarity, 2: We have provided improved justification for the need of optical flow datasets based on multiphase phenomena in Section 4.1. We have also added a section on analysis of the obtained results in Appendix B which further justifies the need for physics-informed optical flow datasets such as BubbleML.
>
> 5. Limitations: We have added a paragraph on limitations in Section 5.

---

> > ### Comment · Reviewer_EMa8 · 2023-08-27
> > **reply**
> >
> > I thank the authors for making the corresponding changes; I raise my score to 7.

---

### Official Review · Reviewer_RFDf · 2023-07-22
**A dataset for phase-change phenomena**

**Rating:** 9
**Confidence:** 4

**Strengths:**

1. The dataset is well designed and the data are carefully processed with different scenarios and parameters.
2. The size of the data is enough for a model in deep learning
3. The dataset may be used to assistant in some benckmark in related areas.


**Additional Feedback:**

No.

**Clarity:**

Yes, the paper is well written and well-organized. The meaning is clear and the structure is good.

**Correctness:**

Their experiments are based on the physic equations and they have provided a clear analysis.

**Documentation:**

Yes. A Github link is provided.  https://github.com/HPCForge/BubbleML

**Ethics:**

No.

**Limitations:**

No. I think there is no obvious limitations.

**Opportunities For Improvement:**

It will be much better if the dataset can contain some real experiment data.

**Relation To Prior Work:**

Yes. The related work has been introduced in due diligence.


**Summary And Contributions:**

The paper introduced a dataset on Phase-change phenomena, which consists of a wide range of boiling phenomena, including nucleate boiling of single bubbles, merging bubbles, flow boiling in different configurations, and subcooled boiling. The data is not from real world experiments but  generated through Flash-X simulations.  To make the simulation real, the dataset covers various gravity conditions ranging from earth gravity to gravity at the International Space Station, different heater temperatures and also different inlet velocities. They also established two benchmark based on this dataset.

In the trends of AI4Science, we need more and more such datasets for training our models and pave the way for further study of ML in various physic problems.

---

> ### Author Response · Authors · 2023-08-22
>
> We thank the reviewer for their valuable comments to improve our paper. We agree that the addition of real world experimental data would make the dataset more comprehensive. However it is very difficult to obtain experimental ground truth for velocity, pressure, and temperature fields and requires extensive collaboration between computer scientists and domain experts (i.e., experimentalists in phase-change physics). We have included the lack of experimental data as a limitation of our work and avenue for future improvements in Section 5. Additionally, we would like to direct your attention to the global response where you can find an overview of the various modifications we have made throughout the paper.

---

### Official Review · Reviewer_uCQg · 2023-07-22
**A benchmark dataset for boiling scenarios and multi-phase dynamics**

**Rating:** 8
**Confidence:** 3
**Correctness:** All fine.

**Strengths:**

* New challenge / datasets for ML of boiling scenarios.
* Baseline results for two challenging down-stream tasks.

**Additional Feedback:**

* Does AWS cloud storage guarantee a long-term availability of the data?
* In the appendix, the reader is referred to zenodo rather than AWS. At the given link, there is, however, only the single-bubble.tar.gz scenario available.
* And for the versioning of the datasets, the authors refer to github in Appendix E.7. These descriptions do not match.
* How was the claimed real-world validation done? Where does the comparison data stem from?

**Clarity:**

The paper is in a good style.

"tranisent behavior" -> "transient behavior"

**Documentation:**

The appendix is required to understand more details.
I would have appreciated further documentation on the website of how to use the datasets without running the baseline ML models.

**Limitations:**

Limitations are not discussed. However, societal impact can not be expected.

**Opportunities For Improvement:**

* The distinction from other scientific ML datasets is a bit misleading. Previous ML datasets are described as too "simple", lacking "the necessary variability to represent the range of behaviors and phenomena encountered in scientific applications". The BubbleML datasets do not represent the claimed range either, which is just too wide. It, however, complements existing SciML datasets by a new scenario that has not been covered by now (at least to the best of my knowledge). Furthermore, the optical flow scenario differs completely from those in the cited references. I recommend to rephrase from claiming that existing datasets are of little use to introducing a novel scenario for benchmarks that has not been considered so far.

* Which of the datasets in Table 1 was used to compute the results in Table 2? (Or in general in Section 4?)

* Highlight the best results in Table 3 in the appendix. Even if this implies to highlight only a single column.

* The link on page 5 to the "dedicated environment [...] provided for running new simulations" is leading to a 404 page not found error: https://github.com/akashdhruv/Multiphase-Simulations

**Relation To Prior Work:**

See above

**Summary And Contributions:**

The paper introduces a dataset of simulations of boiling scenarios. The data differs from existing datasets due to its unique characteristics. It is multi-phase (fluid-vapor) and has unique properties due to its two-phase dynamics (bubbles). With that, the problem of boiling poses difficult challenges for ML: It is time-dependent and of chaotic nature and thus not smooth, and the surfaces and topology of the face separation is difficult to learn.

The dataset is introduced for two downstream ML tasks. First, optical flow analysis to capture the dynamics of bubbles, and second, learning temperature dynamics with operator networks.

The work first motivates the importance of the simulated scenarios, then the need for better (new?) data.
Section 3 introduces the physics of the scenario and the types of datasets that were generated. It furthermore provides a strong indication that the datasets reflect the expected physics. Section 4 then discusses the two, rather different, downstream tasks. The first one, optical flow, requires post-processing of the data into training and validation images. The second one, called SciML, is a more classical scenario for time-dependent simulation data, predicting the next step in time. For the optical flow, the pre-training of an ML model on different external datasets is compared. For the "SciML" results, three different ML approaches (UNet, FNO, and UNO) are compared with respect to

The main contribution is clearly to provide new datasets for boiling scenarios in the Scientific ML realm.

---

> ### Author Response · Authors · 2023-08-22
>
> We thank the reviewer for their valuable comments to improve our paper. We have summarized our responses below in a sequential manner.
>
> 1. Opportunities for improvement, Q.1: We have updated our Related Work Section 2 to rephrase our claims. We believe that the discussion is more fair and complete.
>
> 2. Opportunities for improvement, Q.2: We have added the details of the simulations used to generate the optical flow datasets in the caption for Table 2 and also addressed it in the “Learning bubble dynamics” paragraph of Section 4.1. We have also included results on optical flow datasets generated from all other datasets in Appendix B.
>
> 3. Opportunities for improvement, Q.3: We have highlighted the best results in each table in bold text.
>
> 4. Opportunities for improvement, Q.4: We have corrected the link to the repository and updated it in the footnotes of page 5.
>
> 5. Limitations: We have added a paragraph on limitations in section 5
>
> 6. Clarity: Thank you for pointing out the typo. We have fixed it.
>
> 7. Documentation: We have added Appendix A to address these concerns and also added a set of jupyter notebook examples to the github repository. These show how to load the data, view the relevant fields, visualize the data, and train a neural operator. We have also included documentation discussing how to use the dataset and understand the metadata. These are linked in Appendix A.
>
> 8. Additional Feedback, Q.1: Future maintenance plans is addressed in Appendix A.2
>
> 9. Additional Feedback, Q.2: Zenodo has a limit of 50GB per dataset, which is not enough for our full dataset. We have updated the Zenodo doi with a bash script that downloads all the datasets using “wget”.
>
> 10. Additional Feedback, Q.3: We have updated the website of our dataset to https://hpcforge.github.io/BubbleML/ where we will provide all future versioning information. We have also linked to the website in our paper.
>
> 11. The real world validation is done by comparing our findings with experimental results reported in literature. These papers are cited in the Dataset Validation section. We have also updated the captions of Figures 5 and 6 in the validation section with the appropriate citations so they are more easily accessible.

---

> > ### Comment · Reviewer_uCQg · 2023-08-27
> > **Reply to authors**
> >
> > The suggestions for improvement made by all reviewers have been implemented in detail and carefully. The paper has again clearly improved in quality. In particular, the addition of further ML models has significantly strengthened the benchmark character of the paper. So I am changing my rating from 7 to 8.

---

### Author Response · Authors · 2023-08-22

Dear reviewers,

Thank you for your suggestions on improving the paper. We have made significant changes and enhancements to the paper incorporating your feedback. Changes are marked in blue. In summary, the main changes are as follows:

1. Documentation, Tutorials, and Model Zoo: We have restructured our Github repository to include two example jupyter notebooks that explain the data loading and model training processes and also added documentation of the dataset that describes each of the fields and scalars. We have also included a model zoo in the github repository that includes links to all the trained models and their respective benchmark results.

2. We have added additional details in  Appendix A on how BubbleML conforms to FAIR data principles and justification for the selection of simulation parameters.

3. We have added new benchmarks for scientific machine learning, incorporating many of the suggestions from the reviewers and significantly enhanced the benchmarks. These changes can be found in Section 4.2 and Appendix C.

4. We have included current limitations of the dataset in Section 5.

---

### Decision · Program_Chairs · 2023-09-22

**Decision:**

Accept (Spotlight)

**Comment:**

Spatiotemporal fields generated from direct simulations of different boiling scenarios.
Supported by documentation, tutorials and application examples.
Attractive due to high physical complexity and relevance.